https://doi.org/10.1038/s41467-019-13646-9　**OPEN**

# Forty-five patient-derived xenografts capture the clinical and biological heterogeneity of Wilms tumor

Andrew J. Murphy [1,2]*, Xiang Chen [3], Emilia M. Pinto [4], Justin S. Williams[3], Michael R. Clay [4], Stanley B. Pounds [5], Xueyuan Cao[5,6], Lei Shi[5], Tong Lin[5], Geoffrey Neale [7], Christopher L. Morton[1], Mary A. Woolard[1], Heather L. Mulder [3], Hyea Jin Gil[1], Jerold E. Rehg[4], Catherine A. Billups[5], Matthew L. Harlow[8], Jeffrey S. Dome [9], Peter J. Houghton[10], John Easton[3], Jinghui Zhang [3], Rani E. George[8], Gerard P. Zambetti [4] & Andrew M. Davidoff[1,2]

The lack of model systems has limited the preclinical discovery and testing of therapies for Wilms tumor (WT) patients who have poor outcomes. Herein, we establish 45 heterotopic WT patient-derived xenografts (WTPDX) in CB17 scid$^{-/-}$ mice that capture the biological heterogeneity of Wilms tumor (WT). Among these 45 total WTPDX, 6 from patients with diffuse anaplastic tumors, 9 from patients who experienced disease relapse, and 13 from patients with bilateral disease are included. Early passage WTPDX show evidence of clonal selection, clonal evolution and enrichment of blastemal gene expression. Favorable histology WTPDX are sensitive, whereas unfavorable histology WTPDX are resistant to conventional chemotherapy with vincristine, actinomycin-D, and doxorubicin given singly or in combination. This WTPDX library is a unique scientific resource that retains the spectrum of biological heterogeneity present in WT and provides an essential tool to test targeted therapies for WT patient groups with poor outcomes.

[1] Department of Surgery, St. Jude Children's Research Hospital, 262 Danny Thomas Place, Memphis, TN 38105, USA. [2] Division of Pediatric Surgery, Department of Surgery, University of Tennessee Health Science Center, 910 Madison Ave. 2nd floor, Memphis, TN 38163, USA. [3] Department of Computational Biology, St. Jude Children's Research Hospital, 262 Danny Thomas Place, Memphis, TN 38105, USA. [4] Department of Pathology, St. Jude Children's Research Hospital, 262 Danny Thomas Place, Memphis, TN 38105, USA. [5] Department of Biostatistics, St. Jude Children's Research Hospital, 262 Danny Thomas Place, Memphis, TN 38105, USA. [6] College of Nursing, University of Tennessee Health Science Center, 920 Madison Ave, Memphis, TN 38163, USA. [7] Hartwell Center for Bioinformatics and Biotechnology, St. Jude Children's Research Hospital, 262 Danny Thomas Place, Memphis, TN 38105, USA. [8] Department of Pediatric Hematology and Oncology, Dana-Farber Cancer Institute and Boston Children's Hospital, Harvard Medical School, 450 Brookline Avenue, Room D640E, Boston, MA 02215, USA. [9] Division of Oncology, Children's National Medical Center, 111 Michigan Avenue NW, Washington, DC 20010, USA. [10] Greehey Children's Cancer Research Institute, University of Texas Health Science Center, 8403 Floyd Curl Drive, San Antonio, TX 78229, USA. *email: andrew.murphy@stjude.org

Wilms tumor (WT) is the most common kidney cancer in children and accounts for 6–7% of childhood cancer[1]. Despite the overall success in treatment achieved by large-scale cooperative trials conducted over the last 40 years, patients with unfavorable histology WT (diffuse anaplasia), disease relapse, or bilateral tumors (stage V) continue to have suboptimal outcomes[2].

Diffuse anaplasia occurs in 6% of WT, is associated with resistance to treatment, and accounts for 50% of WT deaths. The development of diffuse anaplasia is likely a late event in tumorigenesis that is related to selection of a TP53-mutant clone within an initially favorable histology WT[3–5]. Historically, tumors with diffuse anaplasia were found to be resistant to vincristine, actinomycin-D, and doxorubicin, chemotherapy drugs which constitute the mainstay of therapy for most favorable histology disease[6].

Approximately 50% of WT patients with favorable histology who experience disease relapse eventually succumb to their disease[7,8]. Key genetic features of WT prone to disease relapse include mutations in TP53, SIX1, or SIX2 with concomitant microRNA-processing gene mutations, and chromosomal copy number alterations (CNAs) including 1q gain and loss of heterozygosity (LOH) of both 1p and 16q[9–12]. For WT treated according to the Children's Oncology Group (COG) very low-risk protocol with surgical resection only, disease relapse was higher in those with LOH or loss of imprinting (LOI) of chromosome 11p15 than those with normal imprinting status at 11p15[13].

For 5–7% of WT patients who present with synchronous bilateral disease (stage V), current treatment protocols mandate neoadjuvant chemotherapy followed by surgical resection, most often in the form of bilateral nephron-sparing surgery. In National WT Study-5, patients with bilateral WT (BWT) had significantly lower event-free survival (EFS; 56%) than those with stage IV (metastatic) unilateral WT (75%); however, the recent prospective study COG AREN0534 has reported substantial improvement in 4-year EFS (82.1%) for BWT patients[14–16]. The rate of end stage renal disease related to treatment of BWT is 14% but can be up to 85% in patients with WT predisposition syndromes (e.g., Beckwith–Wiedemann; Denys–Drash; and WT, aniridia, genitourinary malformation, and range of developmental delays (WAGR) syndromes)[16]. Thus, there is an urgent need to develop models to test alternative treatment strategies for patients with unfavorable histology WT, disease relapse, or BWT with the aim of optimizing cure and minimizing treatment toxicity.

The development and testing of preclinical therapies focusing on these key groups of WT patients have been limited by the paucity of available relevant in vitro and in vivo WT models[17]. The one commercially available, purported WT cell line (WT-CLS1) was recently reclassified as a malignant rhabdoid tumor due to presence of hemizygous SMARCB1 mutation[18]. The G401 cell line was also previously reclassified as a rhabdoid tumor line[19,20]. The previously widely used anaplastic WT cell line, SK-NEP-1 has a gene expression profile characteristic of Ewing sarcoma, including expression of the EWS-FLI1 fusion transcript[20,21]. In addition, although the genetically engineered mouse model of WT (Igf2-Wt1 mice) represents a breakthrough in the understanding of WT genetics, it contains genetic alterations that correspond to a very treatable subset of WT patients and lacks the genetic features of high-risk favorable histology or unfavorable histology disease[22].

Given the deficiency of preclinical models for high-risk WT, in this study we establish heterotopic WT patient-derived xenografts (WTPDX) from 45 patient tumors, including those with diffuse anaplasia (unfavorable histology), disease relapse, or bilateral tumors. The purpose of this study is to provide a practical, focused description of this WTPDX resource to accelerate research for WT

groups with poor outcomes. We determine the similarities and differences in histology, molecular profile, gene expression, and methylation patterns between WTPDX and corresponding primary tumors. Additionally, we show that WTPDX retain the predicted treatment resistance or sensitivity of their histologic subtype (unfavorable vs. favorable histology), thereby offering major opportunities for modeling therapeutic response in WT patients.

## Results

**Establishment and histologic evaluation of WTPDX.** We attempted to establish heterotopic WTPDX from 83 patient tumor samples following WT resection performed at St. Jude Children's Research Hospital (St. Jude) from 2007 to 2016. Of those, 64 (71%) were successfully engrafted. There were no significant differences between primary tumors for which engraftment was successful or failed with respect to patient gender, age at diagnosis, primary resection or neoadjuvant chemotherapy status, histology, disease stage, primary tumor weight, or patient EFS (Supplementary Fig. 1). A total of 47 WTPDX were available for this study (17 initially engrafted xenograft models were lost due to a technical storage issue). Two WTPDX were subsequently excluded because human DNA could not be detected, and immunostaining was consistent with murine T-lymphocytic proliferation (Supplementary Fig. 2).

Corresponding primary tumors were available for histology studies from 45 patients and for molecular analyses from 39 patients. Short tandem repeat (STR) DNA profiling showed a 100% match between WTPDX and corresponding primary tumors (n = 39) (Supplementary Data 1). Included in the total of 45 WTPDX are those established from patient tumors with diffuse anaplasia (n = 6), patients with BWT (n = 13), and patients who later experienced disease relapse (n = 9) with some overlap among these three groups (Fig. 1). All xenografts were established from primary kidney tumor tissue at the time of initial surgical resection with the following three exceptions: KT-23 was established from a pulmonary metastasis in a WT patient with disease relapse and KT-29 and KT-51 were established from local relapse tissue at the time of reoperation. Figure 1 shows clinical characteristics and demographics of patients and matched xenograft models. For the 17 patients who received neoadjuvant chemotherapy, the therapy received and resulting primary tumor response are detailed in Supplementary Data 2.

Histologic concordance was noted between WTPDX and primary tumors in 38 of 45 cases (84.4%; Fig. 2; Supplementary Data 2). Immunostaining revealed retention of critical regulators of nephrogenesis that are highly expressed in WT including blastemal nuclear expression of SIX2 and expression of WT1 in WTPDX (Fig. 2). In addition, large hyperchromatic nuclei with atypical mitotic figures and positive p53 immunostaining in unfavorable histology WT were seen in both the primary tumor and WTPDX (Fig. 2). Histologic discordance (n = 7, 15.6%) included diffuse anaplastic primary tumors with corresponding WTPDX that did not meet the full criteria for diffuse anaplasia (n = 2; KT-23, 71), focal anaplasia in primary tumors but not WTPDX (n = 1; KT-67), triphasic primary tumors that evolved into monomorphic blastemal WTPDX (n = 2; KT-43, 58), or those that evolved into monomorphic stromal or skeletal muscle predominant WTPDX (n = 2; KT-29, 52). NUMA1 immunohistochemical analysis (a human-specific nuclear antigen) in the WTPDX models revealed that the percentage of human-derived cells was 60–95% (mean 90%, median 95%; Supplementary Data 2). The percentage of blastema was found to be significantly higher in WTPDX compared to primary tumors (95% CI 7.9–27.3% higher, paired two-tailed t-test p = 0.0007; Fig. 3; Supplementary Data 2). We examined a focused group of nine

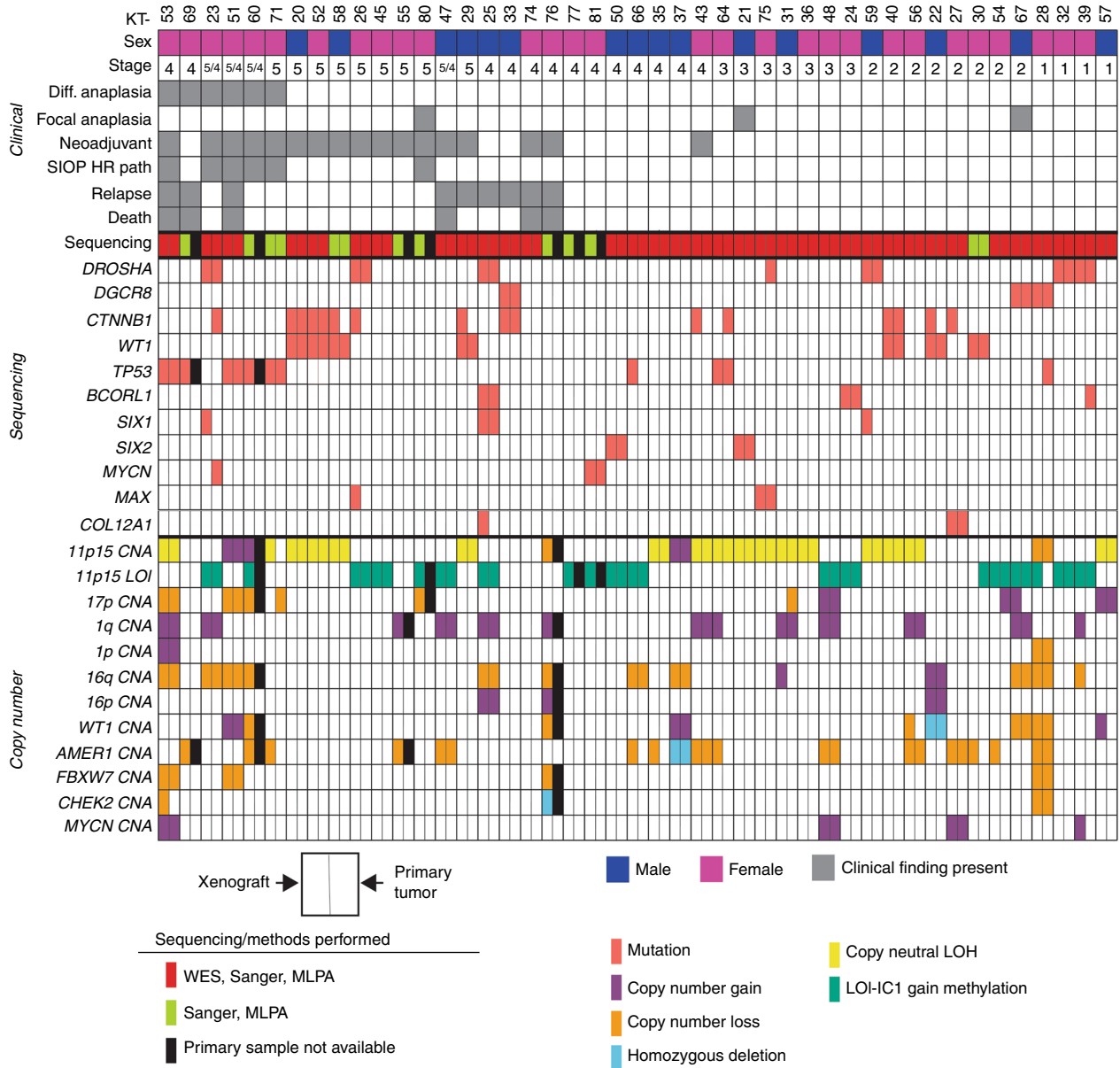

**Fig. 1 45 WTPDX and corresponding primary tumors.** Each KT column represents an individual xenograft. Clinical characteristics are outlined in the top panel. Children's Oncology Group (COG) disease stage is indicated. For bilateral WT cases (stage 5) also with metastasis present, the stage is indicated as 5/4. Neoadjuvant indicates that the patient received neoadjuvant chemotherapy before surgical resection and establishment of the WTPDX. SIOP HR path (as per International Society for Pediatric Oncology Classification) indicates high-risk histology (diffuse anaplasia or blastemal predominance) after neoadjuvant chemotherapy. Relapse indicates that the patient later experienced relapse during the clinical course. Genetic variants and chromosomal copy number alterations (CNAs) are displayed in the lower two panels. The type of sequencing performed (WES with target capture sequencing, Sanger sequencing, or both) is shown. Each finding is depicted for the WTPDX—primary tumor pair, with the left rectangle representing presence of the variant or CNA in the WTPDX, and the right rectangle representing an equivalent finding in the parent primary tumor. An unfilled (white) rectangle indicates absence of this finding in either the WTPDX or primary tumor. Specimens for which primary tumor tissue was not available are indicated by black rectangles.

triphasic primary tumors to determine the change in percent blastema across five xenograft passages. We noted a significant enrichment of blastema by the first passage that was maintained for at least five passages (95% CI 17.4–70.6% higher, paired two-tailed $t$-test $p = 0.0046$; Fig. 3).

**WTPDX capture the genetic heterogeneity of WT.** Single nucleotide variants were profiled by whole exome sequencing (WES) for 35 WTPDX–primary tumor sample pairs with available germline DNA (Fig. 1, Supplementary Fig. 3, Supplementary

Data 3). For the 35 WTPDX–primary tumor pairs that underwent WES, the median number of non-silent single nucleotide variants detected in primary tumors was 7 (IQR 6–9) compared to 5 (IQR 2–8) in WTPDX (paired two-tailed $t$-test $p = 0.0002$; Supplementary Data 3). All 45 WTPDX underwent focused Sanger sequencing and multiplex ligation-dependent probe amplification (MLPA) for chromosomal CNAs (Fig. 1). Of six WTPDX from primary tumors with diffuse anaplasia (KT 23, 51, 53, 60, 69, 71; Fig. 1), five harbored *TP53* mutations (p.R175H, p.R273H, p.R283H, p.R342P, and deletion of exons 4–5), and four had chromosome 17p LOH, consistent with loss of the wild type allele

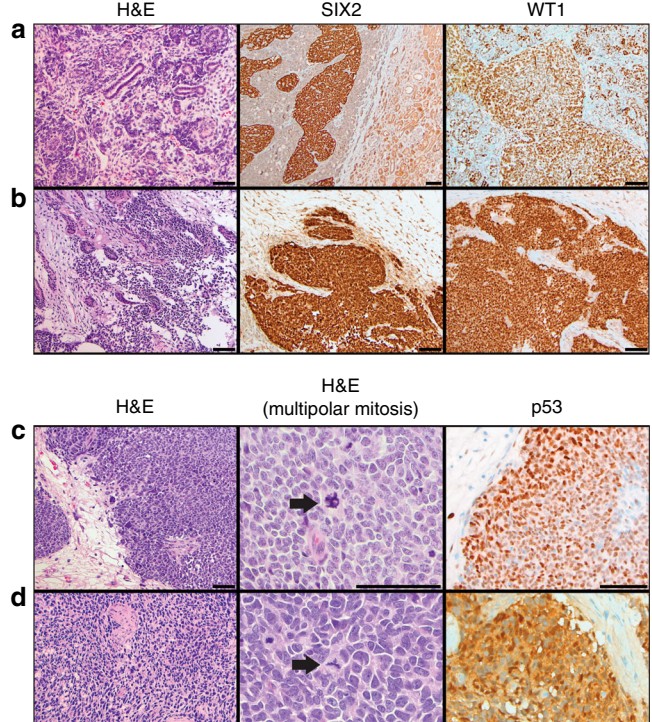

**Fig. 2 WTPDX recapitulate histologic and molecular phenotypes of primary tumors.** Paired favorable-histology, triphasic primary Wilms tumor (WT) **a** and WTPDX **b** corresponding to case KT-45 are shown in the top two rows. WTPDX **b** maintain the epithelial, stromal, and blastemal triphasic histology of primary WT **a**. WTPDX maintain blastemal SIX2 expression (middle) and WT1 expression (right) **b** compared with the primary WT **a**. Paired unfavorable-histology (diffuse anaplasia) WT **c** and WTPDX **d** corresponding to case KT-60. WTPDX **d** maintain the hyperchromatic, large nuclei, and atypical multipolar mitoses (arrows) of the primary WT **c**. WTPDX **d** retain p53 immunostaining that is characteristic of tumors with *TP53* missense mutations. Scale bar = 100 μm.

(Fig. 1). Mutations were detected in both the DNA binding and tetramerization domains of p53 (Supplementary Fig. 4). In addition, two favorable histology WTPDX (KT-64, 66) had heterozygous *TP53* mutations (p.R156C and p.R175H).

The p.Q177R hotspot mutation in either the *SIX1* or *SIX2* transcription factor was found in five WTPDX. Three of these WTPDX (KT-23, 25, 59) were found to have *SIX1/2* p.Q177R hotspot mutations with concomitant p.E1147K hotspot mutations in the microRNA-processing gene *DROSHA* (Fig. 1). The combination of *SIX1/2* p.Q177R and microRNA-processing gene mutations has been previously associated with poor prognosis[10–12]. Overall, nine WTPDX were found to have microRNA-processing gene mutations in either *DROSHA* or *DGCR8* (Fig. 1).

*WT1* mutations with concomitant *CTNNB1* mutations were found to be prevalent in cases of BWT (Fig. 1), consistent with previous reports[23]. Two WTPDX were found to have mutations in *BCORL1* and one WTPDX had a mutation at the p.P44L hotspot in *MYCN* (Fig. 1).

Aberrations at 11p15 were found in 87% of WTPDX (Fig. 1). Methylation-sensitive-MLPA (MS-MLPA) and microsatellite analysis revealed 11p15 copy neutral-LOH with selective loss of the maternal and duplication of the paternal chromosome 11p15 in 17 (37.8%) WTPDX consistent with paternal uniparental disomy (UPD). Two (4.4%) cases (KT-28, 76)

exhibited reduction to hemizygosity with selective loss of the maternal chromosome 11p15. Additionally, 18 (40%) WTPDX exhibited LOI, which was defined by biallelic methylation of H19/IC1 at the 11p15 region. Also, 3 (6.7%) WTPDX showed selective paternal 11p15 copy number gain (KT-37, 51, 60) and 5 (11.1%) WTPDX exhibited normal copy number and methylation status at 11p15 (KT-27, 33, 55, 69, 74; Fig. 1). Of note, chromosomal CNAs and 11p15 imprinting status were consistent between WTPDX and corresponding primary tumors in all cases except KT-71. 13 of 45 (28.9%) WTPDX were found to have chromosome 1q gain (Fig. 1) and 1 WTPDX (2.2%, KT-28) had combined LOH at 1p and 16q.

**WTPDX demonstrate clonal selection and evolution.** Although WTPDX were shown to capture the genetic heterogeneity and multitude of driver mutations found in WT, Fig. 1 depicts many circumstances in which mutations known to be drivers in WT are present in either the primary tumor or xenograft, but not both. We used high coverage target capture sequencing validation data (Supplementary Data 3) to examine subclonal changes between primary tumors and xenografts. Mutant allele frequency plots for WTPDX and primary tumor pairs demonstrated preservation of primary tumor subclones in xenografts, but also detected clonal selection and independent xenograft-specific clonal evolution depending on the individual WTPDX model (Fig. 4).

**WTPDX enrich for a blastemal gene expression pattern.** A Spearman correlation matrix was generated using RNA-seq mean log2(FPKM + 0.01) values and demonstrated highly correlated overall gene expression between paired primary tumors and WTPDX (Fig. 5). To explore potential differences in gene expression, we focused on 9 primary tumor–WTPDX pairs (of 37 total pairs) that constituted the lowest quartile for Spearman correlation (r < 0.836). This low Spearman correlation quartile was associated with neoadjuvant chemotherapy and a histologic transition from blastemal poor primary tumors to enrichment of blastema in corresponding WTPDX (Fig. 5). Using results of a paired two-tailed *t*-test comparing mean log2(FPKM + 0.01) values from WTPDX to primary tumors (Supplementary Data 4), changes in gene pathways were analyzed using the Protein Analysis Through Evolutionary Relationships (PANTHER) classification system database[24]. In the complete cohort, there were no statistically significant pathways upregulated in WTPDX and the integrin-signaling pathway was noted to be significantly downregulated in WTPDX (Benjamini–Hochberg adjusted *p* = 0.001; Supplementary Fig. 5). When analysis was limited to the nine primary tumor–WTPDX pairs that constituted the lowest quartile for Spearman correlation, cell cycle and ubiquitin proteasome pathways were found to be upregulated in WTPDX and PDGF and integrin signaling pathways were found to be downregulated (Supplementary Fig. 5).

Principal component analysis (PCA) of the entire RNA-seq dataset differentiated both primary tumors and WTPDX from normal samples (kidney and commercially available pooled fetal kidney RNA; Fig. 5). Principal component 1 (PC1) explained 21.6% of variance among samples. Exploration of PC1 demonstrated enrichment of WT blastemal archetype genes and kidney developmental cap mesenchyme genes in xenografts compared to primary tumors. This was corroborated by displaying the percent blastema as determined by histology (Fig. 5). Principal component 2 (PC2) explained 10.5% of variance among samples and was associated with enrichment of WT epithelial archetype genes and kidney epithelial genes in WTPDX when compared to primary tumors (Fig. 5)[25].

WTPDX retained the enriched expression of genes previously shown to be upregulated in WT and kidney developmental genes

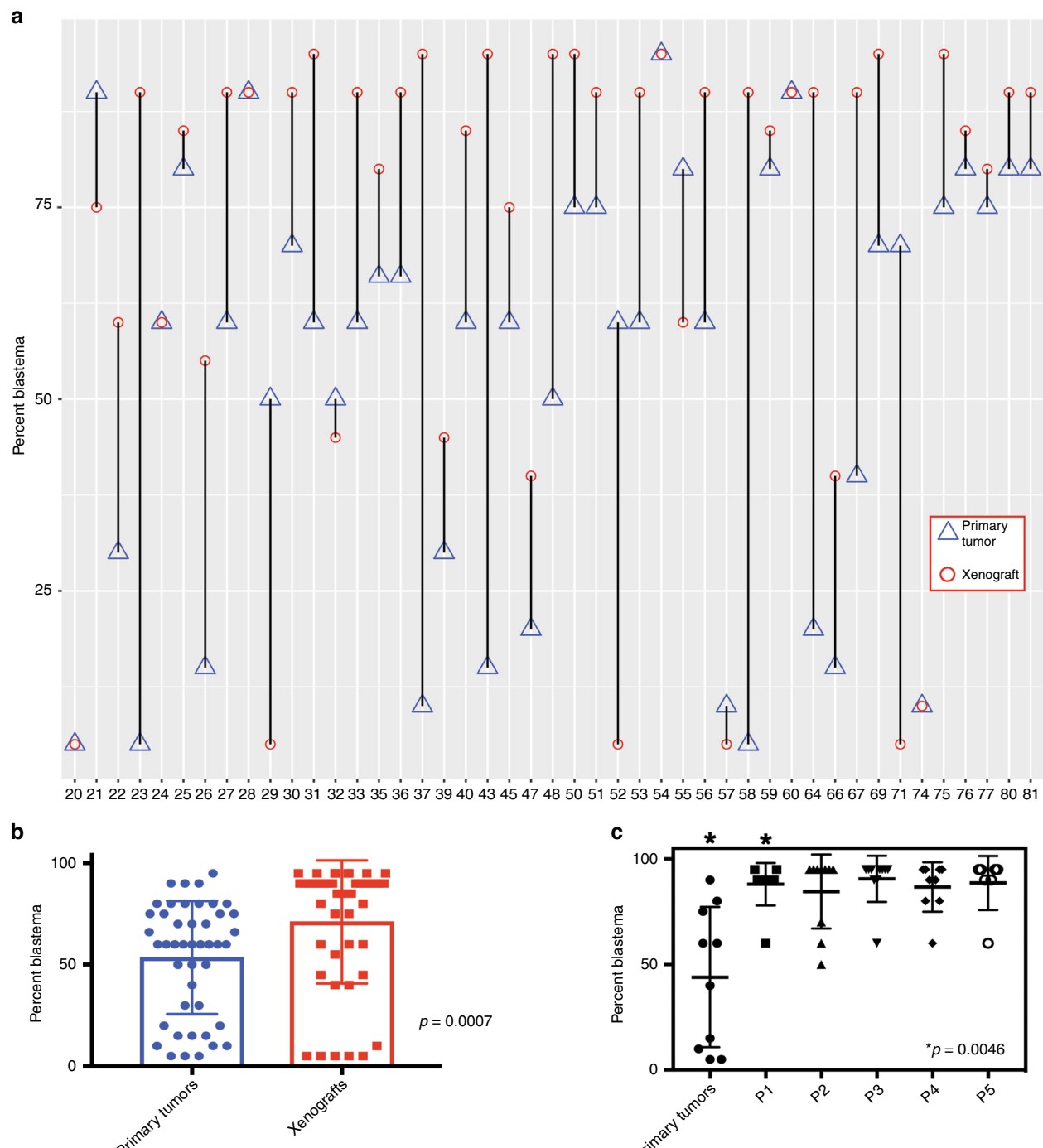

**Fig. 3 Early-passage WTPDX enrich for blastema by histology. a** The percentage of blastema is plotted for primary tumors (triangles) versus corresponding early-passage xenografts (circles), demonstrating that most xenografts enrich for the blastemal component by histology. **b** A significant increase in the percentage of blastema by histology was found when the entire group of primary tumors versus early-passage xenografts were compared. **c** The percent blastema was compared between nine triphasic primary tumors and the first five passages of corresponding WTPDX (P1-P5). A significant increase in percent blastema was found at the first passage and maintained through passage 5. Error bars in **b** and **c** represent mean ± standard deviation. *P* values are from a paired two-tailed *t*-test.

of the renal progenitor cells of the metanephric mesenchyme (Supplementary Fig. 6). In contrast, WTPDX demonstrated reduced expression of immune response genes and vascular endothelial growth factor (VEGF) gene sets (Supplementary Fig. 7). Of note, WTPDX showed negative infiltrating murine stromal and vascular endothelial cells by NUMA1 immunostaining (Supplementary Fig. 8). Gene expression findings were

corroborated using gene expression microarray (Supplementary Figs. 8 and 9, Supplementary Data 5).

**Methylation and global chromosomal copy number analysis.** The retention of methylation status in WTPDX–primary tumor pairs was 70–90% (median 86%) of all methylation sites (Fig. 6).

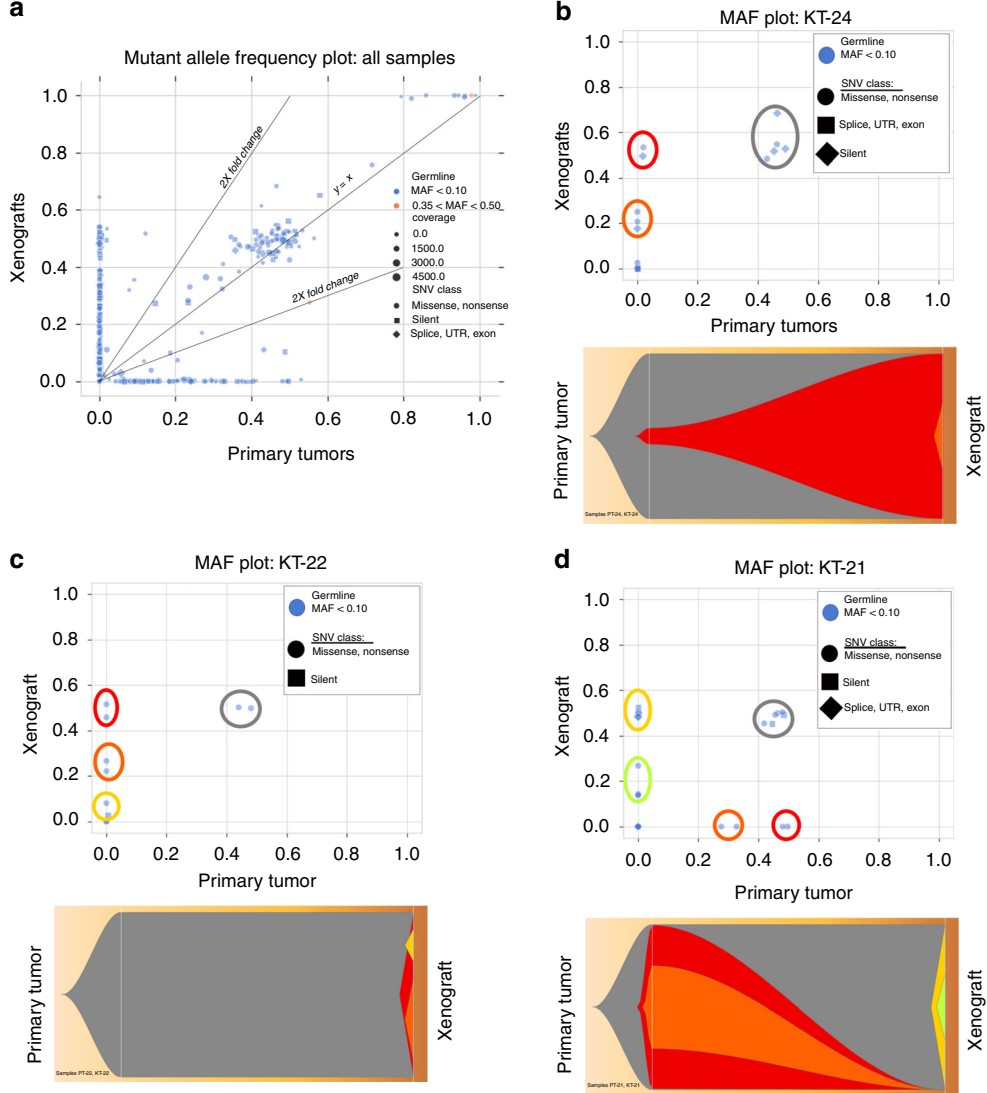

**Fig. 4 Subclonal analysis of WTPDX and parent primary tumors. a** Mutant allele frequency (MAF) plot of the entire target capture validation sequencing dataset depicts clones of mutant alleles shared between WTPDX and primary tumors (along slope $y = x$ line), clones enriched or depleted in the xenografts (2× fold change lines depicted), and clones represented in the primary tumor or xenograft, but not both (clustered along $x$ and $y$ axes). **b** MAF and Fish plots show that KT-24 demonstrates maintenance of a major clone from the associated primary tumor (gray), expansion of a minor subclone from the primary tumor into a major clone in the xenograft (red), and the presence of a xenograft-specific subclone (orange). **c** KT-22 demonstrates maintenance of the major clone from the primary tumor (gray), with emergence of xenograft specific clones (red, orange, and yellow). **d** KT-21 demonstrates maintenance of the major clone from the primary tumor (gray), loss of clones from the primary tumor (red and orange), and emergence of xenograft-specific clones (yellow and green).

Differences found in mean genome-wide DNA methylation ($\beta$ values) between primary tumors and WTPDX were explained by the greater abundance of highly methylated sites and fewer partially methylated sites in primary tumors than in WTPDX. Hierarchical clustering analysis of global methylation data revealed that 31 of 39 tumors clustered together with the corresponding WTPDX (Fig. 6). Of the eight WTPDX that did not cluster together with their corresponding primary tumors by methylation, six (KT-23, 26, 30, 55, 58, 64) models also clustered with the lower Spearman correlation group by RNA-seq analysis (Fig. 5) and five were from tumors pre-treated with neoadjuvant chemotherapy (KT-23, 26, 45, 55, 58). KT-23 and 58 were among the minority of WTPDX with discordant histology when compared with the corresponding primary tumors.

Genome-wide copy number analysis showed that recurrent CNAs present in primary tumors were also maintained in corresponding xenografts (Fig. 6, Supplementary Fig. 10).

However, a subgroup of WTPDX exhibited chromosomal copy number losses in only the WTPDX models but not the primary tumors (Fig. 6, Supplementary Fig. 10).

**Xenograft therapeutic response studies**. Two representative *TP53* mutant anaplastic WTPDX models (KT-51 p.R273H DNA-binding domain mutant and KT-53 p.R342 tetramerization domain mutant) and the favorable histology WTPDX models KT-43, 45, 47, and 75 were treated with vincristine, actinomycin-D, and doxorubicin as monotherapies and in combination for modeling the Children's Oncology Group COG EE4A and DD4A 2-drug and 3-drug chemotherapy regimens (Table 1). Overall, the anaplastic models demonstrated progressive disease to monotherapies and combination therapies; however, KT-53 showed a transient response and subsequent rapid regrowth to regimens containing vincristine

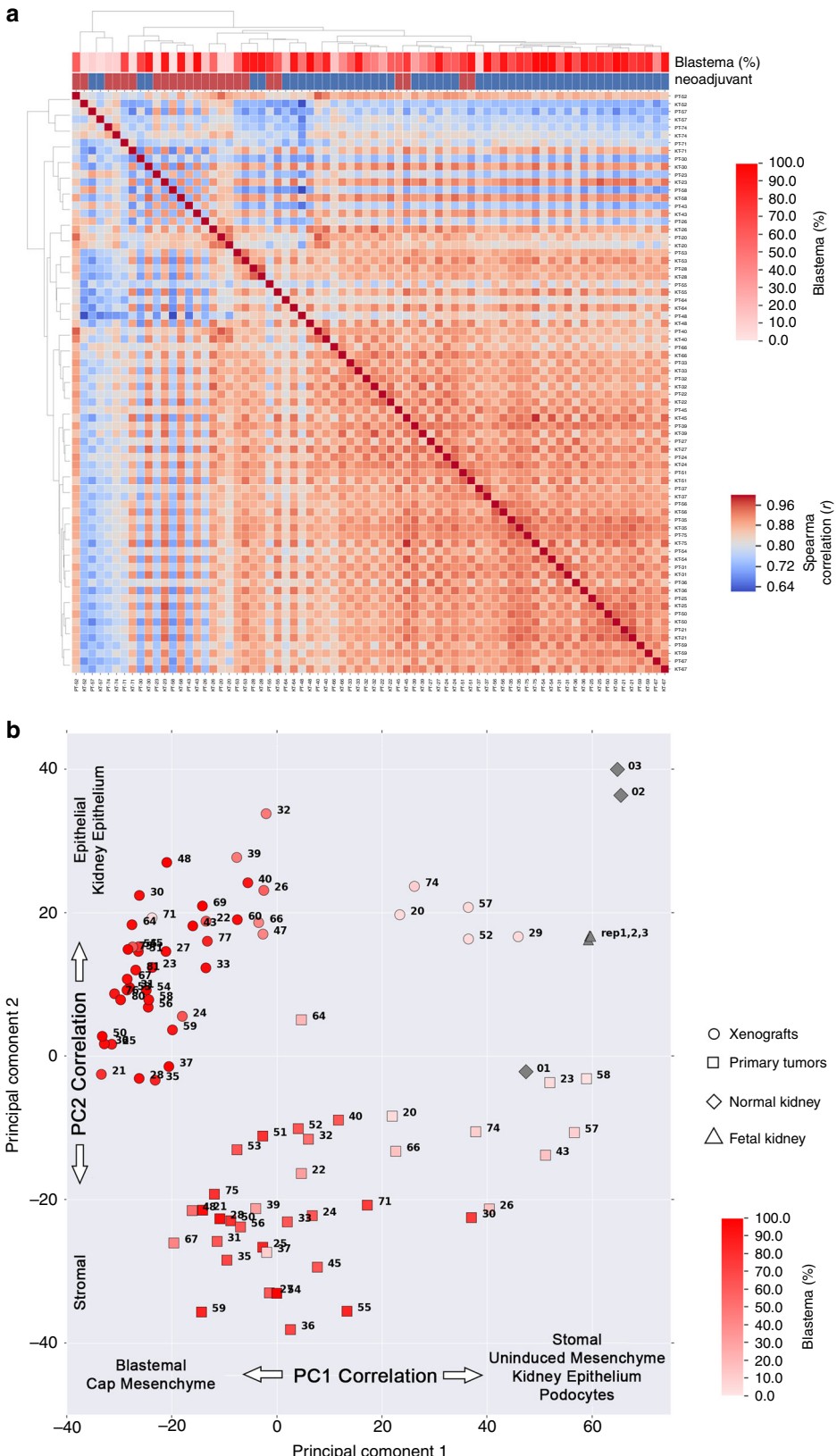

(Fig. 7, Table 1). The favorable histology models showed complete responses or maintained complete responses to vincristine-containing regimens, including the 2-drug and 3-drug chemotherapy regimens (Table 1). Representative growth treatment-response curves are shown in Fig. 7.

## Discussion

Herein, we established an annotated library of 45 heterotopic patient-derived xenografts that captures the clinical and biological heterogeneity of WT. Our diverse panel of WTPDX includes comprehensively annotated in vivo models of unfavorable

**Fig. 5 WTPDX enrich for blastemal gene expression. a** Transcriptomic analysis of RNA-seq data using a Spearman correlation matrix demonstrates highly correlated gene expression between paired WTPDX and primary tumors in most sample pairs. Sample pairs constituting the lowest quartile of Spearman correlation ($r < 0.836$, blue regions of matrix) were associated with neoadjuvant chemotherapy and blastemal poor primary tumors that often transitioned to blastemal-rich WTPDX (arrows). **b** Principal component analysis was performed for paired WTPDX (circles) and primary tumors (squares), normal kidney specimens (diamonds), and pooled fetal kidney RNA (triangles) using RNA-seq data. WTPDX clustered distinctly from primary tumors. PC1 explained 21.6% of variance among samples and was inversely correlated with expression of WT blastemal archetype and kidney development cap mesenchyme genes. This increased expression of blastemal genes in WTPDX corresponded to increased percent blastema detected on histologic analysis (heatmap display of percent blastema per specimen shown). PC2 explained 10.5% of variance among samples and was positively correlated with expression of WT epithelial and kidney epithelial genes. KT—xenografts, PT—primary tumor, NK—normal kidney, FK—pooled fetal kidney RNA.

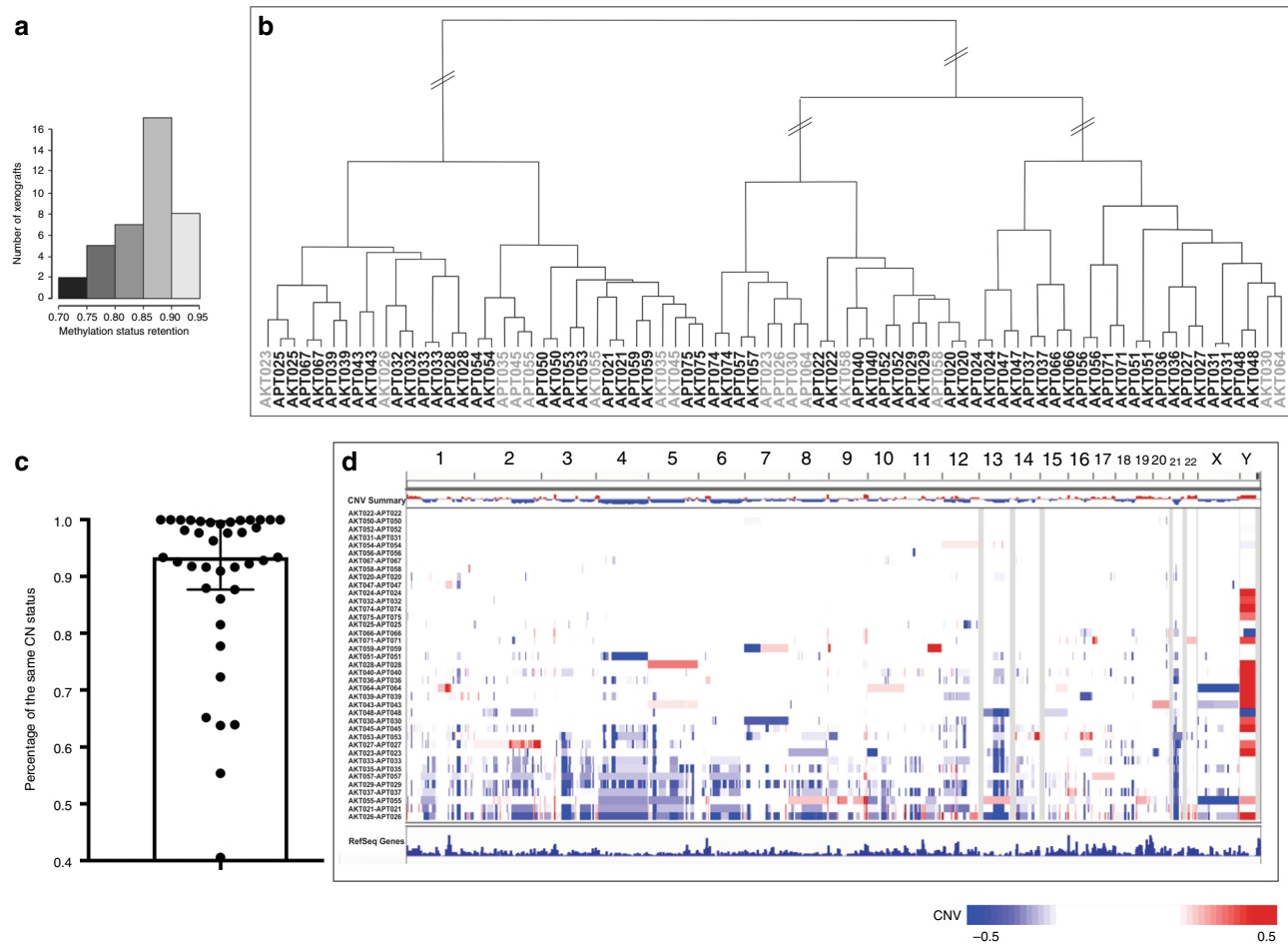

**Fig. 6 Global methylation and chromosomal copy number analysis. a** Methylation retention status of the WTPDX–primary WT pairs ranged from 0.7248 to 0.9167 (median 0.8678). **b** Hierarchical clustering of primary tumors and xenografts according to global methylation status of the top 20,000 most variable CpG site probes in the dataset resulted in the clustering of 31 of 39 xenograft–primary tumor pairs immediately adjacent to each other (matched specimens shaded in black). **c** Boxplot representation (median with tails representing interquartile range) of the percentage of shared copy number status between primary tumors and xenografts. **d** Differential genome-wide chromosomal copy number status comparing primary tumors and xenografts. Each row represents a xenograft–primary tumor pair. Chromosome numbers are given across the top. Blue indicates chromosomal copy number loss in xenografts relative to primary tumors, and red indicates copy number gain. Some xenografts (bottom of graphic) had accumulation of copy number losses (blue) compared with paired primary tumors.

histology WT (*TP53* mutation and/or 17p loss) and models of WT with propensity for disease relapse (combined microRNA-processing gene mutations and *SIX1/2* mutations) that have not been previously captured by genetically engineered mouse models or cell lines. Furthermore, compared with xenograft models established from WT cell lines, our library of WTPDX more closely recapitulates the histologic features (e.g. triphasic histology, blastemal predominance, and anaplasia) of human WT and maintains the expression of key regulators of self-renewing nephron progenitors (SIX2) and regulators of nephrogenesis (WT1)[17]. Therefore, the WTPDX–primary tumor pairs provide a

unique opportunity to study the genetics and biochemistry of human WT and to functionally challenge these findings in this preclinical model.

WT is a malignancy of renal stem cells, and we found that differences in gene expression between primary tumors and WTPDX are partially explained by enrichment for blastemal genes and genes associated with the cap mesenchyme in kidney development. As the blastemal compartment of WT is thought to be the proliferative component of the tumor, we accordingly saw upregulation of cell-cycle genes associated with this phenomenon. Increased blastemal gene expression was correlated

**Table 1 Treatment response of xenografts chemotherapy.**

| Xenograft (histology) | Chemotherapy group | KM estimate median time to event (days)[a] | p value[b] | EFS T/C (days)[c] | Median RTV at end of study[d] | Tumor volume T/C (cm³)[e] | p value[f] | Median group response |
|---|---|---|---|---|---|---|---|---|
| KT-51 (UH) | V | 17.36 | 0.282 | 1.3 | 99.00 | 0.68 | 0.005 | PD1 |
| KT-51 (UH) | A | 11.98 | 0.126 | 0.9 | 99.00 | 1.38 | 0.038 | PD1 |
| KT-51 (UH) | D | 14.12 | 0.501 | 1.0 | 99.00 | 1.31 | 0.195 | PD1 |
| KT-51 (UH) | VA | 17.05 | 0.394 | 1.2 | 11.19 | 1.42 | 0.072 | PD1 |
| KT-51 (UH) | VAD | 21.54 | 0.073 | 1.6 | 58.74 | 0.56 | 0.005 | PD1 |
| KT-53 (UH) | V | 49.55 | <0.001 | 4.1 | >10 | 0.02 | <0.001 | CR |
| KT-53 (UH) | A | 12.31 | 0.192 | 1.0 | >10 | 0.89 | 0.234 | PD1 |
| KT-53 (UH) | D | 12.01 | 0.853 | 1.0 | >10 | 0.80 | 0.040 | PD1 |
| KT-53 (UH) | VA | 39.02 | <0.001 | 3.2 | >10 | 0.04 | <0.001 | PD2 |
| KT-53 (UH) | VAD | 40.05 | 0.006 | 3.3 | >10 | 0.05 | 0.012 | PD2 |
| KT-43 (FH) | V | – | 0.002 | >3.8 | 8.9649 | 0.15 | 0.002 | CR |
| KT-43 (FH) | A | 24.20 | 0.997 | 1.1 | 16.4177 | 1.24 | 0.876 | PD1 |
| KT-43 (FH) | D | – | – | – | – | – | – | – |
| KT-43 (FH) | VA | – | 0.002 | >3.8 | 54.6278 | 0.21 | 0.017 | CR |
| KT-43 (FH) | VAD | – | 0.002 | >3.8 | 1.9462 | 0.09 | 0.008 | CR |
| KT-45 (FH) | V | – | 0.001 | >3.4 | 0 | 0.00 | <0.001 | MCR |
| KT-45 (FH) | A | – | 0.001 | >3.4 | 0.5257 | 0.02 | <0.001 | CR |
| KT-45 (FH) | D | 47.23 | 0.004 | 1.9 | 99 | 0.29 | <0.001 | PD2 |
| KT-45 (FH) | VA | – | 0.001 | >3.4 | 0 | 0.00 | <0.001 | MCR |
| KT-45 (FH) | VAD | – | 0.001 | >3.3 | 0 | 0.00 | <0.001 | MCR |
| KT-47 (FH) | V | >EP | <0.001 | >4.9 | 0.39 | 0.01 | <0.001 | MCR |
| KT-47 (FH) | A | 41.43 | <0.001 | 2.4 | >10 | 0.16 | <0.001 | PR |
| KT-47 (FH) | D | 24.80 | <0.001 | 1.5 | >10 | 0.21 | <0.001 | PD1 |
| KT-47 (FH) | VA | >EP | 0.002 | >5.0 | 0.20 | 0.01 | 0.004 | MCR |
| KT-47 (FH) | VAD | >EP | <0.001 | >4.9 | 0.24 | 0.01 | <0.001 | CR |
| KT-75 (FH) | V | – | <0.001 | >4.7 | 0 | 0.00 | <0.001 | MCR |
| KT-75 (FH) | A | 29.60 | <0.001 | 1.6 | 99 | 0.28 | <0.001 | PD1 |
| KT-75 (FH) | D | – | – | – | – | – | – | – |
| KT-75 (FH) | VA | – | <0.001 | >4.7 | 0 | 0.00 | <0.001 | MCR |
| KT-75 (FH) | VAD | – | <0.001 | >4.7 | 0 | 0.00 | 0.001 | MCR |

PD1—progressive disease 1 (>25% increase in tumor volume with tumor growth delay value ≤1.5)
PD2—progressive disease 2 (>25% increase in tumor volume with tumor growth delay value >1.5)
CR—complete response—disappearance of measurable tumor mass (<0.10 cm³) for at least one time point
MCR—maintained complete response—tumor mass <0.10 cm³ at end of study period
PR—partial response—tumor volume regression of ≥50% for at least one time point but with measurable tumor (≥0.10 cm³)
*FH* favorable histology, *UH* unfavorable histology (diffuse anaplasia), *V* vincristine, *A* actinomycin-D, *D* doxorubicin, *VA* vincristine + actinomycin-D, *VAD* vincristine + actinomycin-D + doxorubicin, *EP* evaluation period, *EFS* event-free survival, *KM* Kaplan–Meier, *RTV* relative tumor volume
[a]Kaplan–Meier estimate of median days to event was determined by using interpolated days to event. >EP indicates that median event free survival for the treated group is greater than the evaluation period (EP)
[b]Exact log-rank test *p* values comparing event free survival distributions between treated and control groups
[c]EFS T/C: ratio of the median time to event between treated and control groups
[d]Median final RTV is the ratio of tumor volume at the end of treatment to that at initiation of treatment
[e]Tumor volume T/C is the ratio of mean tumor volume of treated tumors divided by that of control tumors
[f]Wilcoxon rank sum test *p* values comparing tumor volumes between control and treated groups

with a histologic observation of increased percent blastema in most WTPDX when compared to their parent primary tumors. When looking at a selection of triphasic primary tumors, it appears this enrichment for blastema occurs during the process of engraftment and persists across passages. Garvin et al. originally observed enrichment for blastema with serial passaging of WT xenografts[26]. Dekel et al. previously showed that accumulation of blastemal elements in serially passaged WT xenografts originally derived from a single patient tumor was associated with increased cellular proliferation, upregulation of cell cycle genes, and upregulated expression of imprinted/paternally expressed genes associated with differentiation blockade and maintenance of cellular self-renewal[27]. This group also later showed that serial passaging of WT xenografts was associated with promoter hypomethylation and upregulated expression of renal progenitor genes including *WT1*, *PAX2*, and *SALL1*[28]. In favorable histology, serially passaged xenografts, from multiple patient sources, this enrichment of nephron progenitor genes that promote stemness was exploited to expand and identify WT cancer initiating cells/cancer stem cells marked by NCAM and later, more specifically, combined NCAM/ALDH1 expression[29–31]. NCAM-targeted

therapies were used to selectively deplete the cancer stem cell population in WT and this approach may prove to be an effective therapeutic strategy for relapsed or refractory disease[32]. Trink et al. developed computational archetypes corresponding to the blastemal, epithelial, and stromal cellular populations of WT. They found that WT of advanced stage exhibited gene expression clustering with the blastemal archetype[25]. However, a comprehensive genomic characterization of each xenograft model utilized in these prior studies was not performed and therefore these resources may be of limited utility to new preclinical studies. Our broad analysis in a large number of xenograft models confirms that WTPDX exhibit increased expression of blastemal archetype genes when compared with their parent primary tumors and therefore may represent a model of WT progression.

We observed reduced expression of genes associated with the immune response, extracellular matrix, and vascular endothelium in WTPDX when compared to primary tumors. This could in part be explained by the immunodeficient environment in CB17 scid$^{-/-}$ mice and the murine-derived tumor stroma and neovascularity. Molecular characterization of tumors by the Pediatric Preclinical Testing Program found the greatest

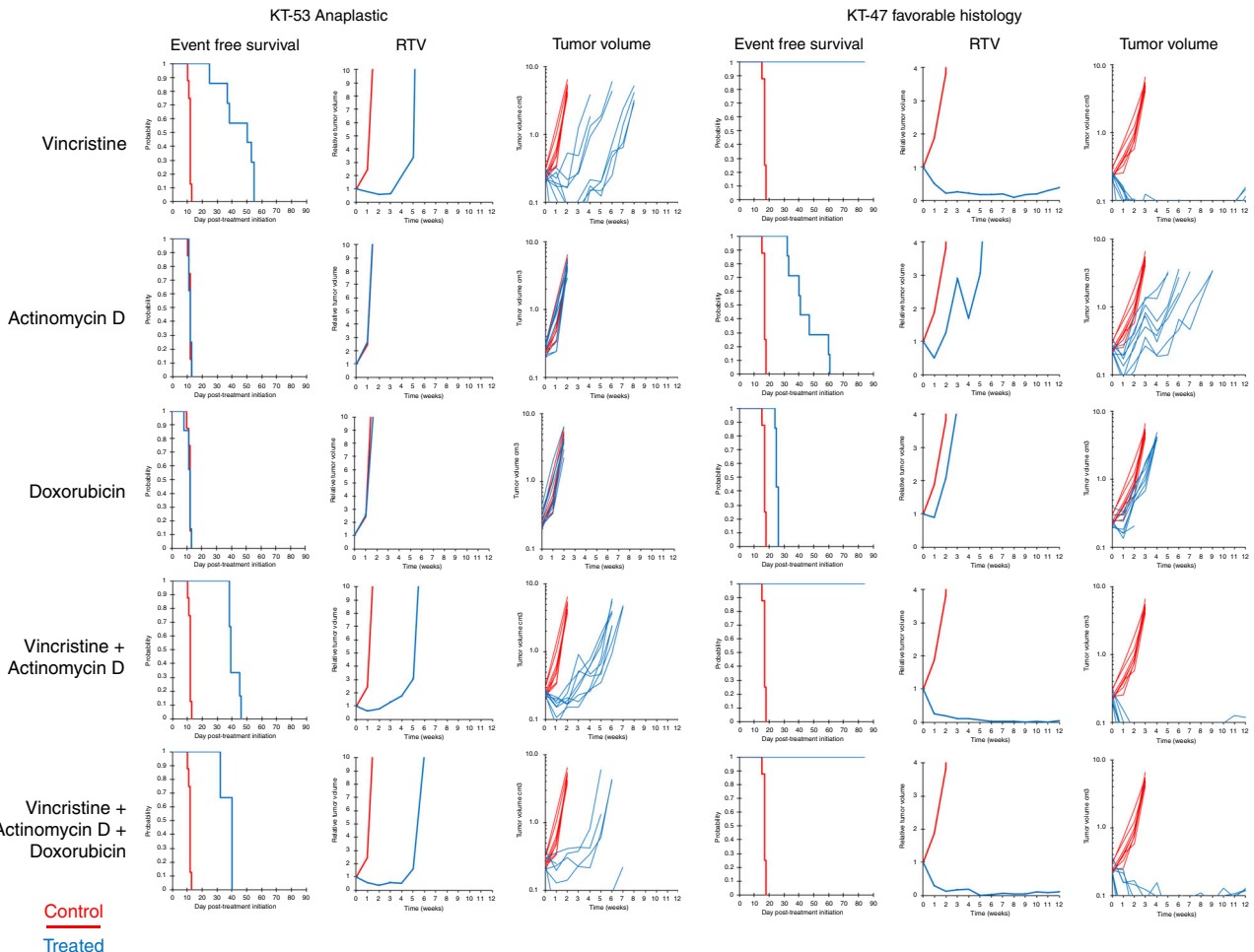

**Fig. 7 Treatment responses in anaplastic versus favorable-histology WTPDX.** The Children's Oncology Group (COG) EE-4A regimen is modeled by the vincristine and actinomycin-D 2-drug combination treatment, and the COG DD-4A regimen is modeled by the vincristine, actinomycin-D, and doxorubicin 3-drug combination treatment. RTV—median relative tumor volume (cm$^3$).

difference in gene expression between xenografts and primary tumors was due to reduced expression of endothelial and vascular genes[33]. Additionally, Garvin et al. observed that reduced extracellular matrix in the xenografts was the only ultrastructural cellular difference between a primary WT and serially passaged WT xenografts[26].

We found the highest concordance between WTPDX and their parent primary tumors was for LOH or LOI at chromosome 11p15. In this WTPDX library, LOH or LOI at 11p15 was present in 87% of tumors. This is slightly higher than LOH or LOI rates of 70–80% reported previously for WT and may be due to the high proportion of BWT in this WTPDX library. Despite finding widespread intra-tumor heterogeneity involving several chromosomal loci, Cresswell et al. found little to no WT intra-tumor heterogeneity at 11p15[34]. These findings suggest that 11p15 LOH or LOI is the most common, unifying early event in WT tumorigenesis and are additionally supported by the Wt1-Igf2 mouse model and by the propensity of patients with Beckwith–Wiedemann syndrome (germline UPD or LOI at 11p15) to develop WT[22]. We also noted high concordance between WTPDX and paired primary tumors for variants in the microRNA-processing genes *DROSHA* and *DGCR8* and the renal progenitor transcriptional regulators *SIX1* and *SIX2*, adding additional support to their role as early, cooperating events in Wilms tumorigenesis[10–12].

Our results also show differences in the number of variants detected by WES when comparing WTPDX to primary tumors. Overall, a higher number of genetic variants were found in WTPDX compared to primary tumors. This could be due to subclonal selection in the process of xenograft engraftment, a sampling phenomenon due to intra-tumor genetic heterogeneity known to be present in WT, continued tumor evolution in the xenograft model, or likely all the above. Our subclonal analysis demonstrates specific instances of subclonal selection and tumor evolution in the xenograft model. WT has previously been shown to have significant intra-tumor genetic heterogeneity, which may allow for clonal expansion of more aggressive tumor phenotypes[34,35]. Consistent with this concept of clonal selection and evolution within xenograft models, a recent study evaluating a large solid tumor patient-derived xenograft library reported a higher number of mutations and chromosomal CNAs in xenografts than in corresponding primary tumors[36].

Chemotherapy response assays show that WTPDX phenocopy predicted tumor response by capturing the treatment resistance of *TP53* mutant unfavorable histology WT and the relative sensitivity of favorable histology disease to 2-drug or 3-drug combination therapy with vincristine, actinomycin-D, and doxorubicin.

In summary, WTPDX are a biologically and clinically relevant model system that capture the histology, diverse genetic drivers, epigenetic characteristics, and chemotherapy response of primary WT. Our library of 45 WTPDX will serve as a unique, freely

available scientific resource to conduct preclinical treatment studies and accelerate WT research for patients with diffuse anaplastic tumors, disease relapse, and bilateral tumors.

## Methods

**Ethics statement**. All experiments were conducted in accordance with St. Jude Institutional Review Board-approved protocols (NR08-062, XPD16-126, and XPD17-017). Animal studies were conducted in accordance with the protocol approved by the St. Jude Institutional Animal Care and Use Committee (protocol 031).

**Establishment of heterotopic WTPDX**. Fresh primary samples of human WT were transplanted subcutaneously onto the flanks of male CB17 $scid^{-/-}$ mice (6–8 weeks old; Taconic Farms, Hudson, NY, USA) as previously described[37]. Initial tissue transplants typically grew within 2 months and were maintained in vivo by subsequent serial passages into healthy mice. No cell culture intermediates were used. Early passage (1–3) xenograft tumor tissue was snap frozen in liquid nitrogen for molecular studies, and an adjacent fragment was fixed in 10% neutral buffered formalin for histologic and immunohistochemical studies. Second passage (P2) WTPDX material was used for all molecular analyses except P1 was used for KT-20 and P3 for KT-64 due to specimen availability.

**Authentication of WTPDX by STR analysis**. Genomic DNA was extracted from 45 WTPDX and 39 available corresponding primary tumors by using the QIAamp DNA mini kit (Qiagen, Hilden, Germany). Genetic profiling analysis included 15 STR analysis loci plus Amelogenin included in the PowerPlex® 16 System kit (Promega, Madison, WI, USA). Electropherograms for all multiplex PCR-amplified products were obtained by using the 3730 xl DNA Analyzer (Applied Biosystems, Foster City, CA, USA) and analyzed by using GeneMapper software (Thermo-Fisher Scientific, Waltham, MA, USA).

**Histologic analysis**. Histologic analysis was conducted by a pediatric pathologist who was blinded to clinical and molecular data. All histology and immunohistochemical analyses were performed using an Olympus BX46 microscope (Olympus, Tokyo, Japan) and images were captured using an SC180 camera and CellSens Standard Software v1.18 (Olympus). White balancing, resizing, and cropping were performed using Adobe Photoshop v13.0 (Adobe, San Jose, CA, USA). Hematoxylin-and-eosin-stained sections from formalin-fixed, paraffin-embedded tissues of 45 WT and corresponding WTPDX were reviewed to characterize tumor histology. The percentage of blastema was calculated by reviewing all available tissue sections from primary tumors and corresponding WTPDX. WTPDX sections were compared with the primary tumor to assess histologic concordance and discordance rates. Histologic discordance was defined as (1) primary tumors with focal or diffuse anaplasia that was not detected in the corresponding WTPDX, (2) stromal predominant, pretreated primary tumors that generated blastemal predominant or triphasic xenografts, or (3) triphasic primary tumors that evolved into monomorphic WTPDX representing only 1 of 3 histologic compartments (blastemal, stromal, or epithelial).

**Immunohistochemical analysis**. Immunohistochemical analysis was performed on 4-μm deparaffinized tissue sections from primary and WTPDX tumors by using Benchmark XT (Ventana Medical, Oro Valley, AZ, USA) and BondMax (Leica Microsystems, Wetzlar, Germany) automated stainers. Primary antibodies for p53 (1:200 dilution, Zeta Corp, Sierra Madre, CA, USA), WT1 (1:25 dilution, Agilent Technologies, Santa Clara, CA, USA), SIX2 (1:50 dilution, ProteinTech, Chicago, IL, USA), NUMA1 (1:75 dilution, LifeSpan Biosciences, Seattle, WA, USA), and mouse CD3 (1:1000 dilution, Santa Cruz Biotechnology, Dallas, TX, USA) were used according to manufacturers' recommendations. Appropriate positive and negative controls were included. Staining was classified as negative, weak, moderate, or strong, and cellular localization as nuclear, cytoplasmic, or both. Immunohistochemical detection of the human cell-specific nuclear antigen NUMA1 was used to determine the proportion of human (positive) and murine-derived (negative) cells per WTPDX as previously described[38–42].

**Whole exome sequencing**. WES was possible for 35 WTPDX–primary tumor samples pairs that also had available germline DNA. Detailed methods are available in the Supplementary Methods. Sequencing was performed on an Illumina NovaSeq 6000 instrument using paired 100 cycle dual indexed chemistry. This protocol was followed for all sample types, including germline, primary tumor, and xenografts (indicated in sample IDs as G, D, or X, respectively).

WES mapping, coverage and quality assessment, single-nucleotide variant (SNV) and insertion/deletion detection, tier annotation for sequence mutations, and prediction of the deleterious effects of missense mutations were performed as previously described[43]. Single nucleotide and insertion/deletion variants were validated by target capture amplicon sequencing using the MiSeq platform (Illumina) and the Validation Capture pipeline as previously described[44] and/or Sanger sequencing with manual analysis (see below). For WTPDX WES and

targeted capture sequencing, the XenoCP method was used to remove murine reads misaligned to the human genome[45].

**Focused sequencing for TP53, WT1, CTNNB1, SIX1, SIX2, and N-MYC**. Because not all WTPDX had available paired primary tumor and germline DNA required for our WES pipeline and because WES may miss microdeletions or mutations in repetitive or GC-rich areas[46], additional focused mutational screening was performed for the entire coding region and intron–exon boundaries of TP53 and WT1 by PCR and Sanger sequencing (3730 xl DNA Analyzer, Applied Biosystems). CTNNB1 exon 3, the Q177R hotspot mutation in SIX1 and SIX2 and the P44L hotspot mutation in N-MYC were also analyzed by Sanger sequencing. All sequence plots were compared to the Genome Reference Consortium Human Build 38 (GRCh38). Primer sequences and PCR conditions are given in Supplementary Data 6.

**Chromosomal copy number analysis**. CNAs for chromosomes 1 and 16, WT1, AMER1, N-MYC, FBXW7, and the TP53 loci were analyzed by MLPA using the SALSA P380 WT MLPA kit (MRC Holland, Amsterdam, The Netherlands). Chromosome 17p CNAs (including the TP53 locus) were examined by using the SALSA P056 P53 MLPA kit (MRC Holland). The GeneMapper Software (ThermoFisher Scientific) was used to perform DNA sizing and allele calls, and plots were generated using Coffalyser software (MRC Holland). Reference samples [normal human kidney (n = 3; Amsbio, Abingdon, United Kingdom) and blood (n = 2; Human Genomic DNA, Promega, WI) from healthy individuals] were included.

**11p15 copy number and methylation analysis**. Chromosomal CNAs and methylation status at H19 (Imprinting center 1, IC1) and KvDMR (IC2) regions adjoining the IGF2 locus at chromosome 11p15 were analyzed in WTPDX and primary tumor samples by MS-MLPA using the SALSA ME030 BWS/RSS MS-MLPA kit (MRC Holland) and Hha I restriction endonuclease (Promega). Also, five microsatellite markers located at 11p15 (D11S1363, D11S922, D11S4046, HUMTH01, and D11S988) were genotyped to detect and analyze allelic losses in genomic DNA from xenograft tumors. LOH was considered if a homozygous pattern was observed for all markers as previously reported[47].

**RNA-sequencing and gene expression microarray analysis**. Total RNA was extracted from 37 primary tumors and WTPDX, eight additional WTPDX without available matched primary tumor RNA, and three normal kidney specimens by using the Qiagen RNeasy Midi kit (Qiagen). Commercially available pooled total RNA from four human fetal kidney specimens was also included (Takara, Kusatsu, Japan). A detailed description of RNA quantification and quality control is available in the Supplementary Methods. RNA-seq library preparation, sequencing, read mapping, and generation of gene level read counts and Fragments per kilobase million (FPKM) values were generated as previously described and detailed in the Supplementary Methods[43]. For all RNA-seq-based analyses, FPKM values were transformed by log2(FPKM + 0.01); all genes with max(log2(FPKM + 0.01)) < 1 were excluded. Generation of the Spearman correlation matrix, gene list analysis, PCA, and normalized z-score heatmap analysis of gene lists of interest are detailed in the Supplementary Methods. The Human Clariom S gene expression microarray (ThermoFisher) was used to corroborate RNA-seq findings using a separate assay as detailed in the Supplementary Methods.

**Global methylation and copy number analysis**. Genomic DNA (1 μg) from 39 WTPDX and corresponding primary tumors was bisulfite converted by using the EZ DNA Methylation kit (Zymo Research Corp, Irvine, CA, USA) according to the manufacturer's instructions. Converted samples were processed and hybridized to the Infinium MethylationEPIC BeadChip (850K) system (Illumina, San Diego, CA, USA) according to published protocols[48]. The methylation score of each CpG site in the array is represented as a beta (β) value and was computed using the methylation module of the GenomeStudio software (version 1.9.0; Illumina).

Methylation array data were normalized by using the subset within-array normalization (SWAN) method as implemented in the minfi Bioconductor package. For each methylation array, genomic copy number analysis of methylation profiles was performed by applying the circular binary segmentation method (DNAcopy package)[49,50] to the total hybridization signal that was adjusted for probe type and GC content with a regression tree[51]. Segmented results were re-centered by the median of segment means at the probe-set level. Segments with a mean <−0.2 and >+0.2 were inferred to be regions with fewer and >2 copies, respectively.

Methylation retention analysis was performed as follows. First, for each array profile, the methylation status of each probe's beta value was defined as unmethylated (β < 1/3), hemi-methylated (1/3 < β < 2/3), or methylated (β > 2/3). For each xenograft–primary tumor pair, the methylation status of a probe was retained if it had the same methylation status in both samples. For each xenograft–primary tumor pair, the methylation retention rate was defined as the proportion of probes that retained their methylation status. Finally, descriptive statistics for the methylation retention rate were computed. Heatmaps and

dendrograms for expression and methylation profiles were built for data visualization by R software.

**Xenograft therapeutic response assays.** Representative anaplastic/unfavorable histology WTPDX (KT-51 and 53) and favorable histology WTPDX (KT-43, 45, 47, and 75) models were selected to determine response to conventional WT chemotherapy treatment. Tumor-bearing mice were treated with actinomycin-D (0.30 mg/kg intraperitoneal q21 days × 3; Baxter Oncology, Lebanon, NJ), vincristine (1 mg/kg intraperitoneal q7 days × 8; Hospira, Lake Forest, IL), and doxorubicin (4 mg/kg IV q7 days × 8; Pfizer, New York, NY, USA), or their combination to model COG EE-4A and DD-4A 2 and 3-drug WT chemotherapy[1]. Mice received the drug(s) when tumor size was between 200 and 500 mm$^3$ as previously described[37,52]. Mice were randomized to groups of eight. Tumor volumes were measured for each tumor at study initiation and weekly for up to 84 days after study initiation or until tumor size reached 2.5 cm$^3$. Assuming that tumors were spherical, tumor volumes were calculated by using the formula $(\pi/6) \times d^3$, where $d$ represents the mean diameter. Tumor status (e.g., progressive disease, relative tumor volumes, and EFS) was determined as previously reported and detailed in the Supplementary Methods[37,52,53].

**Reporting summary.** Further information on research design is available in the Nature Research Reporting Summary linked to this article.

## Data availability

The WES and RNA-seq data are available in the European Genome-phenome Archive (EGA) database (https://www.ebi.ac.uk/ega/home) under accession number EGAS00001003361. The RNA expression array data and methylation profiling array data that support the findings of this study are available in the Gene Expression Omnibus (GEO) database (https://www.ncbi.nlm.nih.gov/geo/) under identifiers GSE110696 (gene expression) and GSE110697 (methylation). All WTPDX used in this study are available to the scientific community upon request. The source data underlying Fig. 1 are summarized in Supplementary Data File 3 and Supplementary Figs. 3 and 4. The source data underlying Figs. 2 and 3 are summarized in Supplementary File 2. The source data for Figs. 4–6 (WES, RNA-seq, and methylation data) have been uploaded to the databases noted above. The source data for Fig. 7 is also summarized in Table 1. A reporting summary for this article is available as a Supplementary Information file.

## Code availability

Custom codes are available from the authors upon request

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

## Acknowledgements

This work was supported by the American Lebanese Syrian Associated Charities (ALSAC/St. Jude Children's Research Hospital), the American Pediatric Surgical Association Foundation Jay Grosfeld, MD Scholarship (A.J.M.), NCI Cancer Center Support Grant CA21765, Team Path to the Cure Grant (R.E.G.), and NCI U01 grant CA199297 (P.J.H.). The authors thank Samantha Spencer, Karen Rakestraw, Emily Walker, Melanie Lloyd, Scott Olsen, Granger Ridout, Dana Roeber, and the Hartwell Center for Bioinformatics and Biotechnology at St. Jude for their technical contributions to this manuscript. We also thank Raven Holcomb and the Department of Pathology for their help with the histologic and immunohistochemical studies. We thank Vani Shanker for her review of the manuscript. Finally, we would like to thank Charles Mullighan, Jeffrey Rubnitz, Matthew Lear, the Tissue Resources Committee, and the St. Jude Biorepository for their assistance with identifying and obtaining the primary tumor specimens.

## Author contributions

Conception and design: A.J.M., X.C., E.M.P., G.N., C.L.M., M.L.H., J.S.D., P.J.H., R.E.G., G.P.Z., and A.M.D.; Development of methodology: A.J.M., X.C, J.S.W., E.M.P., M.R.C., S.B.P., G.N., C.L.M., J.S.D., P.J.H., J.E., J.Z., R.E.G., G.P.Z., A.M.D.; Acquisition of data: A.J.M., X.C., J.S.W., E.M.P., M.R.C., G.N., C.L.M., M.L.H., M.A.W., H.L.M., H.J.G.; Analysis and interpretation of data: A.J.M., X.C., J.S.W., E.M.P., M.R.C., S.B.P., X.C., L.S., T.L., G.N., C.L.M., C.A.B., J.E.R., R.E.G., G.P.Z., A.M.D.; Writing, review, and/or revision of manuscript: A.J.M., X.C., J.S.W., E.M.P., M.R.C., S.B.P., X.C., L.S., T.L., G.N., C.L.M., M.L.H., H.J.G., J.E.R., C.A.B., J.S.D., P.J.H., J.E., J.Z., R.E.G., G.P.Z., and A.M.D.

## Competing interests

The authors declare no competing interests.
