## [Peer Review File · Nature Communications]

Reviewers' comments:

Reviewer #1 (Expertise: PDX models, wilms tumor, therapy, Remarks to the Author):

This is a very interesting paper that shows in large scale (different Wt subtypes) and in a more comprehensive and detailed matter what other groups have already shown about the validity of the human Wilms Tumor xenografts PDX. Unfortunately and surprisingly these studies, which have given insights into WT biology and renal development as well as suggested therapeutic directions are not at all referenced (can be easily traced by inserting wilms tumor xenograft in the PUBMED)

While this WT xenograft collection clearly stand out as a useful resource the work remains rather descriptive. Specifically, novel insights to tumor biology and/or clinical prognostic factors and/or new treatments are lacking.

Reviewer #2 (Expertise: WT genetics, Remarks to the Author):

Whilst I congratulate the authors on this important effort to generate cell lines of Wilms' tumour, the characterisation of the lines is, unfortunately, rudimentary. Without a detailed characterisation of the cell lines, they are not particularly useful in this day and age. Furthermore, the paper is not well written.

MAJOR POINTS

1) The genomic analysis of xenografts utilised outdated methods despite this study coming from one of the world's finest childhood genomics centers. You need to perform whole genome DNA sequencing of xenograft, primary tumours, normal tissue DNA from the child (eg, blood) plus next generation RNA sequencing of all xenografts and primary tumours. As you say yourself, WT has a relatively low somatic mutation burden and therefore, whole exome sequencing will not capture sufficient data points. Whole genome sequencing is required to capture enough data points to be able to deduce the clonal origin, composition and stability of the cell lines, and to capture all possible driver events. Without this data, your effort remains quarter-baked and is not of sufficient gravitas for a journal of such a high calibre.

2) Can you confirm that the cell lines will be made available to researchers around the globe, upon request? If so, just state this somewhere clearly in the manuscript. If not, it would preclude your paper from publication.

3) It is interesting, and very odd indeed, that the xenografts with anaplasia did not respond to triple chemo with Vinc / AD / Dox. From table 1, I gather that 9 patient had anaplasia and received pre-treatment. Did these patients respond to pre-treatment? If so, how come their xenografts were treatment resistant. Similarly, in SIOP practice, anaplastic tumours do often respond to neoadjuvant chemotherapy. So there is a discrepancy between real life children with anaplastic tumours (who can respond) and your unresponsive xenografts. This needs to be explained and questions the validity of your models.

INTERMEDIATE POINTS:

4) This is an invalid, misleading conclusion: "Comparison of these genetic and molecular alterations in primary tumors versus corresponding WTPDX suggests that these genetic events are largely maintained in WTPDX and may be selected for clonal expansion during xenograft engraftment." Looking at Figure 1, there are 117 genomic events listed with paired data on primary and xenograft. Of these, 23 (~20%) are discrepant between tumour and primary. How does a 20% difference translate into "largely maintained"?

5) This is sloppy science: "WT is characterized by high intra-tumor genetic heterogeneity, which may allow for clonal expansion of more aggressive tumor phenotypes." The evidence you quote here is a pretty rudimentary paper that used outdated 300k copy number arrays without interrogating point mutations... Based on this we cannot conclude that WT is characterized by high intra-tumour heterogeneity. Besides, even if the study were valid, what does "high" actually mean? Have you used a quantitative measure that differentiates between "high" and "low"? Have you performed a systematic survey of tumour heterogeneity across human cancer in which WT ranks highly?

5) More sloppiness: "Changes in chromosomal copy numbers, somatic alterations, and epigenetic modifications impose a proportional change in the transcriptome and methylome, possibly favoring the clonal expansion that contributes to the progression of malignancy." How do you know that these modifications impose a proportional change? Have you measured this?

6) Even more sloppiness: "In addition, chemotherapy may accelerate tumor heterogeneity and evolution, as evident in our finding that 5 of 8 WTPDX-primary tumor pairs that did not cluster together by methylation were derived from patients receiving neoadjuvant chemotherapy." The p-

value of this difference must be somewhere in the region of 1. If you want to comment on this, then compare methylation profiles of treatment naïve biopsies to methylation profiles of tumour resections after pre-treatment.

7) In Figure 1, this is implausible: KT 71 has lost 17p in the tumour whereas there is normal copy number of 17p in the xenograft. How do you explain that? (Presumably the explanation is that the xenograft is derived from a subclone within the tumour that is 17p wildtype. If you had whole genome data, you could dig into this...)

8) This is odd: “Two WTPDX were subsequently excluded because they did not faithfully represent the corresponding human tumor (Short tandem repeat [STR] DNA profiling could not amplify human DNA, and histology was not consistent with human WT).” So what were these? Were they tumours? If so, what kind of tumours?

9) The introduction fails to mention the fundamental difference of WT management between the US COG approach (upfront nephrectomy in most children) and the European SIOP approach (neoadjuvant chemotherapy in all cases bar infants < 6 months of age). As such, this category of children essentially does not exist in Europe: “In very low-risk WT patients treated with surgical resection only...”.

MINOR POINTS

10) Did you draw Figure 1 in Excel? It looks a little unprofessional. Please redraw professionally.

11) This is a terrible paragraph: “We opined that each WTPDX would recapitulate the histology, molecular profile, gene expression, and methylation pattern of its corresponding primary tumor. Additionally, we expected that WTPDX would retain the predicted treatment resistance or sensitivity of their histologic subtype (unfavorable vs. favorable histology), thereby offering major 93 opportunities for modelling therapeutic response in WT patients.” This is not hypothesis driven research so why don't you just say that you wanted to make cell lines.

12) The language is imprecise and / or too wordy in places.

- Line 49: “of all pediatric cancer cases” – change to “of childhood cancer”

- Line 50/51: “patients with diffuse anaplasia” – patients do not have diffuse anaplasia; their tumours have.

- Also, patients are children in the first instance, then patients. So why not use children instead of patients?

Reviewer #3 (Expertise: Pediatric tumors, Wilms tumor, therapy, Remarks to the Author):

The paper deals with the clinical and biological heterogeneity of WT that is captured by patients-derived xenografts. The analysis is based on 45 xenografts from 28 patients.

1. The cohort of patients is 28, the number of xenografts from this 28 patients is 45. It would be of interest from which patients more than one xenograft was established and analysed. In addition were there differences found in the histology and/or the molecular findings between different xenografts from the same tumor of a single patient. If so, please discuss how such a heterogeneity in WT will influence new treatment options, if different results from xenografts in one tumor can be obtained.

2. If there is a comparison between the molecular findings and the histology between the xenografts and the primary tumor the following question needs to be answered Is the histology of the primary tumor taken from the same specimen that was used for the xenograft? In other words a biopsy was taken from the primary tumor for the xenograft and was this biopsy splitted into 2 parts one for the xenograft and the other one for the histology and molecular findings? Is this meant by tumor pairs? If this is not the case, can this be a reason for discrepancies between histologies, finding diffuse anaplasia in the primary tumor but not in the xenograft? This question is also important related to line 405 - 408. For the neoadjuvant treated tumors and the discrepancy between primary tumor and xenograft the question would be is the xenograft showing more malignant clones, meaning blastemal subtypes or anaplasia?

Minor point:

line 124: after neoadjuvant chemotherapy high risk blastemal tumors are described as blastemal subtypes. Blastemal predominant tumors are only named in this way after primary surgery, to distinguish them from each other. This is of relevance, as in line 142, it is unclear, if this blastemal predominant tumor was primarily operated or not.

Re: Revised submission of manuscript (NCOMMS-18-04822) to *Nature Communications: Forty-five patient derived xenografts capture the clinical and biological heterogeneity of Wilms tumor*

Dear Reviewers:

After completing all the requested revisions based on reviewer comments and the editor response to our initial rebuttal, I am pleased to resubmit the above-titled manuscript to be considered for publication as an *Article* in *Nature Communications*.

The current revised submission addresses the reviewer concerns by including the following key additional analyses/experiments:

- All models with paired germline, primary tumor, and xenograft DNA have now been characterized by whole exome sequencing with target capture deep sequencing validation.
- A subclonal analysis has been completed using the target capture deep sequencing data and demonstrated subclonal selection, independent clonal evolution, and maintenance of subclones in Wilms tumor xenografts depending on the model system investigated.
- Xenografts and paired primary tumors have been analyzed by total strand RNA-sequencing.
- Analysis of the RNA-seq data demonstrated enrichment of blastemal gene expression that was corroborated by our previous histologic analysis.
- Additional in vivo chemotherapeutic assays with vincristine, actinomycin-D, doxorubicin, and clinically relevant combinations have been performed on a total of 2 anaplastic xenograft lines and 4 favorable histology xenograft lines.

This manuscript has not been published in part or in entirety and is not under consideration by another journal. All studies were conducted under the appropriate institutional review board and animal care and use committee-approved protocols. All of the authors have approved the manuscript and agree with submission to your journal. There are no conflicts of interest to declare. Genome-wide data have been submitted to and accepted by the European Genome-phenome Archive (EGA) and Gene Expression Omnibus (GEO) databases as outlined in the data availability statement and reporting summary.

We believe that this freely available, important scientific resource warrants publication in *Nature Communications* because it stands to significantly accelerate research for Wilms tumor patient groups with suboptimal outcomes if it reaches a broad audience. We believe this publication represents a resource of significant importance to specialists in the field of translational pediatric oncology research and scientists who study pediatric renal tumors. We have worked extremely hard to complete these revisions to your satisfaction because we believe in the importance of this resource, which took 10 years to assemble at a center dedicated to pediatric oncology research.

Please find our itemized responses to the reviewer comments below.

Sincerely,

Andrew Jackson Murphy, MD
Assistant Member, Department of Surgery
St. Jude Children's Research Hospital

Reviewers'

comments:

Reviewer #1 (Expertise: PDX models, wilms tumor, therapy, Remarks to the Author):

This is a very interesting paper that shows in large scale (different Wt subtypes) and in a more comprehensive and detailed matter what other groups have already shown about the validity of the human Wilms Tumor xenografts PDX. Unfortunately and surprisingly these studies, which have given insights into WT biology and renal development as well as suggested therapeutic directions are not at all referenced (can be easily traced by inserting wilms tumor xenograft in the PUBMED)

Response: We have added the following references to the revised version of the manuscript which support the findings of our study that Wilms tumor xenografts enrich for blastemal histology and gene expression:

Garvin, A. J. et al. The in vitro growth, heterotransplantation, and immunohistochemical characterization of the blastemal component of Wilms' tumor. *Am J Pathol* 129, 353-363 (1987). *This study shows that serial passaging of a Wilms tumor xenograft line results in enrichment of blastema and that the xenograft line showed loss of extracellular matrix when compared with the primary tumor by ultrastructural analysis.*

Shukrun, R. et al. Wilms' tumor blastemal stem cells dedifferentiate to propagate the tumor bulk. *Stem Cell Reports* 3, 24-33, doi:10.1016/j.stemcr.2014.05.013 (2014). *This study used serial passaging of Wilms tumor xenografts to enrich for a blastemal stem cell population.*

Pode-Shakked, N. et al. Developmental tumorigenesis: NCAM as a putative marker for the malignant renal stem/progenitor cell population. *J Cell Mol Med* 13, 1792-1808 (2009). *This study used serial passaging of Wilms tumor xenografts to enrich for a blastemal stem cell population.*

Trink, A. *et al.* Geometry of Gene Expression Space of Wilms' Tumors From Human Patients. *Neoplasia* 20, 871-881, doi:10.1016/j.neo.2018.06.006 (2018). This study developed “archetypes” of Wilms tumor gene expression corresponding to blastemal, epithelial, and stromal cell types. They noted that Wilms tumors of higher stages clustered near the blastemal archetype and that Wilms tumor xenografts also clustered together in this blastemal archetype region.

While this WT xenograft collection clearly stand out as a useful resource the work remains rather descriptive. Specifically, novel insights to tumor biology and/or clinical prognostic factors and/or new treatments are lacking.

Response: Rather than to arrive at insights regarding novel treatments, the main purpose of this study is to comprehensively characterize a resource which will accelerate research for Wilms tumor patient groups with diffuse anaplastic tumors, disease relapse, and bilateral disease. We spent the last decade assembling this resource and now aim to characterize and distribute it to the scientific community. While prior studies have utilized Wilms tumor xenografts to test new therapies, the xenograft models in these other studies are not sufficient in number to reflect the genetic heterogeneity of Wilms tumor and are not comprehensively characterized. Therefore, it has not been possible to understand key features of each xenograft line including clinical details, histology, or genetic variants or copy number variants of interest from prior studies. These details are imperative to understand when considering the response to novel therapies. Moreover, the merits of this model system have not been evaluated such that the scientific community understands the strengths and weaknesses of patient-derived xenografts in the context of this specific tumor type. Because Wilms tumor has significant intratumor genetic heterogeneity and because it also has notable genetic heterogeneity in terms of driver mutations contributing to tumorigenesis, we feel that a resource manuscript such as this is imperative in moving the field forward. Nevertheless, the following novel insights into tumor biology are described in the revised version of this manuscript:

1. Wilms tumor patient-derived xenografts enrich for blastemal histology by the first passage.
2. Differential gene expression and genome wide methylation patterns between Wilms tumor patient-derived xenografts and their parent primary tumors is partially explained by an enrichment for blastemal gene expression in xenografts, particularly if they come from tumors that have received neoadjuvant chemotherapy.
3. Genetic variants can differ between Wilms tumor patient-derived xenografts and parent primary tumors because of selection for minor subclones and/or xenograft-specific tumor evolution.
4. Anaplastic xenografts exhibit significant treatment resistance to conventional chemotherapy, offering a major opportunity for testing novel agents and to explore drug resistance in this model system. Previous studies testing a limited number of anaplastic WT xenografts did not test drug combinations used in Wilms tumor therapies.
5. Of the copy number alterations and genetic variants examined, only 11p15 status was uniformly preserved between primary tumors and xenografts. This provides additional evidence that 11p15 loss of heterozygosity or loss of imprinting is an

early event in Wilms tumorigenesis and calls attention to 11p15 alterations as an area of potential therapeutic interest in this and other model systems.

Reviewer #2 (Expertise: WT genetics, Remarks to the Author):

Whilst I congratulate the authors on this important effort to generate cell lines of Wilms' tumour, the characterisation of the lines is, unfortunately, rudimentary. Without a detailed characterisation of the cell lines, they are not particularly useful in this day and age. Furthermore, the paper is not well written.

Response: We made a concerted effort to make it abundantly clear in the introduction and methods sections that this is not a manuscript detailing the generation of cell lines. The patient-derived heterotopic xenografts generated in this study do not come from a cell line intermediate and the experiments are performed on xenograft tumor and primary tumor tissue and not cell lines. Regarding the quality of the writing, I feel that this is an extremely subjective and biased opinion of our manuscript. The manuscript was reviewed and revised by an independent scientific editor prior to submission to *Nature Communications*.

MAJOR

POINTS

1) The genomic analysis of xenografts utilised outdated methods despite this study coming from one of the world's finest childhood genomics centers. You need to perform whole genome DNA sequencing of xenograft, primary tumours, normal tissue DNA from the child (eg, blood) plus next generation RNA sequencing of all xenografts and primary tumours. As you say yourself, WT has a relatively low somatic mutation burden and therefore, whole exome sequencing will not capture sufficient data points. Whole genome sequencing is required to capture enough data points to be able to deduce the clonal origin, composition and stability of the cell lines, and to capture all possible driver events. Without this data, your effort remains quarter-baked and is not of sufficient gravitas for a journal of such a high calibre.

Response: We have revised the manuscript using the methods suggested by the reviewer (except we chose whole exome sequencing rather than whole genome sequencing since this provided sufficient data for a subclonal analysis and because the purpose of this study was not to identify novel noncoding variants). The revised version of this manuscript contains data based on whole exome sequencing with target capture sequencing validation. These methods, including the high depth of coverage of target capture sequencing, enabled a detailed subclonal analysis. The transcriptomic analysis has been redone using total strand RNA-seq.

2) Can you confirm that the cell lines will be made available to researchers around the globe, upon request? If so, just state this somewhere clearly in the manuscript. If not, it would preclude your paper from publication.

Response: The xenografts are freely available to the scientific community. This has been clarified in the manuscript, added to the data availability statement, and included in the reporting summary.

3) It is interesting, and very odd indeed, that the xenografts with anaplasia did not respond to triple chemo with Vinc / AD / Dox. From table 1, I gather that 9 patient had anaplasia and received

pre-treatment. Did these patients respond to pre-treatment? If so, how come their xenografts were treatment resistant. Similarly, in SIOP practice, anaplastic tumours do often respond to neoadjuvant chemotherapy. So there is a discrepancy between real life children with anaplastic tumours (who can respond) and your unresponsive xenografts. This needs to be explained and questions the validity of your models.

Response: Anaplastic tumors that receive vincristine, actinomycin-D, and doxorubicin are known clinically to be treatment resistant to the drugs utilized in our study as established in the early National Wilms Tumor Studies. A reference to these results has been included in the introduction section of the manuscript. Furthermore, the reason an open biopsy has been mandated at 6 weeks of treatment with vincristine, actinomycin-D, and doxorubicin for unresponsive bilateral tumors in the COG AREN0534 study is because this could be indicative of diffuse anaplasia. Therefore, I think our results that anaplastic xenografts are resistant to vincristine, actinomycin-D, and doxorubicin and favorable histology xenografts are sensitive are consistent with what one would expect. To validate this finding using additional xenograft models, we treated a total of two anaplastic xenograft lines (KT-51 and 53) and four favorable histology lines (KT-43, 45, 47, 75) with vincristine, actinomycin-D, and doxorubicin (as monotherapies and in clinically relevant combinations with 8 mice treated per group). Patient responses to neoadjuvant chemotherapy using RECIST criteria have been included in Supplementary File 2.

INTERMEDIATE

POINTS:

4) This is an invalid, misleading conclusion: "Comparison of these genetic and molecular alterations in primary tumors versus corresponding WTPDX suggests that these genetic events are largely maintained in WTPDX and may be selected for clonal expansion during xenograft engraftment." Looking at Figure 1, there are 117 genomic events listed with paired data on primary and xenograft. Of these, 23 (~20%) are discrepant between tumour and primary. How does a 20% difference translate into "largely maintained"?

Response: This language has been eliminated in the revised version of the manuscript. We have reframed our discussion and conclusions to reflect not only that xenografts capture the multitude of genetic drivers that are present in Wilms tumor, but also that xenografts select for subclones of the primary tumor or evolve xenograft specific subclones. Overall, we have revised the manuscript to reflect a more transparent comparison between xenografts and their parent primary tumors.

5) This is sloppy science: "WT is characterized by high intra-tumor genetic heterogeneity, which may allow for clonal expansion of more aggressive tumor phenotypes." The evidence you quote here is a pretty rudimentary paper that used outdated 300k copy number arrays without interrogating point mutations... Based on this we cannot conclude that WT is characterized by high intra-tumour heterogeneity. Besides, even if the study were valid, what does "high" actually mean? Have you used a quantitative measure that differentiates between "high" and "low"? Have you performed a systematic survey of tumour heterogeneity across human cancer in which WT ranks highly?

Response: This language has been eliminated from the manuscript. We have included a phrase in the discussion that states "WT has previously been shown to have significant intra-tumor genetic heterogeneity." Since the initial submission of this manuscript, a landmark manuscript detailing clonal evolution in pediatric solid tumors has confirmed that Wilms tumor contains significant intratumor genetic heterogeneity. Furthermore, this

manuscript also states that accelerated clonal evolution is associated with anaplasia in Wilms tumor and is found in a higher percentage of Wilms tumors than other pediatric solid tumors. This reference has been added to support our statement. Karlsson, J. *et al.* Four evolutionary trajectories underlie genetic intratumoral variation in childhood cancer. *Nat Genet* 50, 944-950, doi:10.1038/s41588-018-0131-y (2018).

5) *More sloppiness: "Changes in chromosomal copy numbers, somatic alterations, and epigenetic modifications impose a proportional change in the transcriptome and methylome, possibly favoring the clonal expansion that contributes to the progression of malignancy." How do you know that these modifications impose a proportional change? Have you measured this?*

Response: This language has been eliminated from the revised manuscript.

6) *Even more sloppiness: "In addition, chemotherapy may accelerate tumor heterogeneity and evolution, as evident in our finding that 5 of 8 WTPDX-primary tumor pairs that did not cluster together by methylation were derived from patients receiving neoadjuvant chemotherapy." The p-value of this difference must be somewhere in the region of 1. If you want to comment on this, then compare methylation profiles of treatment naïve biopsies to methylation profiles of tumour resections after pre-treatment.*

Response: This language has been eliminated from the revised manuscript.

7) *In Figure 1, this is implausible: KT 71 has lost 17p in the tumour whereas there is normal copy number of 17p in the xenograft. How do you explain that? (Presumably the explanation is that the xenograft is derived from a subclone within the tumour that is 17p wildtype. If you had whole genome data, you could dig into this...)*

Response: As depicted in Figure 4 of the revised manuscript, xenografts can: maintain major subclones present in the primary tumors, expand minor subclones present in the primary tumors, lose subclones present in the primary tumors, and develop xenograft-specific subclones. This may be due to subclonal selection from the primary tumor material or spatial heterogeneity in the primary tumor material.

8) *This is odd: "Two WTPDX were subsequently excluded because they did not faithfully represent the corresponding human tumor (Short tandem repeat [STR] DNA profiling could not amplify human DNA, and histology was not consistent with human WT)." So what were these? Were they tumours? If so, what kind of tumours?*

Response: Additional analysis performed on these eliminated xenograft models showed that the small round blue cells in these models represent a murine t lymphocytic proliferation (Supplementary Figure 1).

9) *The introduction fails to mention the fundamental difference of WT management between the US COG approach (upfront nephrectomy in most children) and the European SIOP approach (neoadjuvant chemotherapy in all cases bar infants < 6 months of age). As such, this category of children essentially does not exist in Europe: "In very low-risk WT patients treated with surgical resection only..."*

Response: This project comes from a cohort of patients treated in the United States under COG protocols, and thus mentioning the very-low-risk cohort is relevant as it pertains to

11p15 status. I did not feel that it was an appropriate use of space in this paper to rehash the differences between COG and SIOP protocols.

MINOR

POINTS

10) *Did you draw Figure 1 in Excel? It looks a little unprofessional. Please redraw professionally.*

Response: Figure 1 has been completely redone to have a more professional appearance and to contain data from additional analyses.

11) *This is a terrible paragraph: “We opined that each WTPDX would recapitulate the histology, molecular profile, gene expression, and methylation pattern of its corresponding primary tumor. Additionally, we expected that WTPDX would retain the predicted treatment resistance or sensitivity of their histologic subtype (unfavorable vs. favorable histology), thereby offering major 93 opportunities for modelling therapeutic response in WT patients.” This is not hypothesis driven research so why don’t you just say that you wanted to make cell lines.*

Response: We have changed this language to state: “We sought to determine the similarities and differences in histology, molecular profile, gene expression, and methylation patterns between WTPDX and parent primary tumors. Additionally, we questioned whether WTPDX would retain the predicted treatment resistance or sensitivity of their histologic subtype (unfavorable vs. favorable histology), thereby offering major opportunities for modelling therapeutic response in WT patients.” We feel that these changes more accurately represent the aims of the project.

12) *The language is imprecise and / or too wordy in places.*
- *Line 49: “of all pediatric cancer cases” – change to “of childhood cancer”*
- *Line 50/51: “patients with diffuse anaplasia” – patients do not have diffuse anaplasia; their tumours have.*
- *Also, patients are children in the first instance, then patients. So why not use children instead of patients?*

Response: These changes have been made throughout the manuscript. All the patients in this study are children, so these terms are used interchangeably throughout the manuscript.

Reviewer #3 (Expertise: Pediatric tumors, Wilms tumor, therapy, Remarks to the Author):

The paper deals with the clinical and biological heterogeneity of WT that is captured by patient-derived xenografts. The analysis is based on 45 xenografts from 28 patients.

Response: The methods section clearly details that the xenografts are derived from 45 different cases of tumor resection. The supplementary files containing STR profiling data show a unique identity for each xenograft line. The title of the paper states that there are 45 patient-derived xenografts. For bilateral tumors, only one of the tumors was submitted for xenografting at the time of resection.

1. The cohort of patients is 28, the number of xenografts from this 28 patients is 45. It would be of interest from which patients more than one xenograft was established and analysed. In addition were there differences found in the histology and/or the molecular findings between different xenografts from the same tumor of a single patient. If so, please discuss how such a heterogeneity

in WT will influence new treatment options, if different results from xenografts in one tumor can be obtained.

Response: Xenografts and primary tumor specimens used for molecular analysis were established from similar areas of the tumor submitted for research purposes. Multiple xenografts were not established from the same patient in this study. All xenografts come from patients with a unique identity. In several cases, multiple passages of each xenograft line were compared for histology. We observed a general phenomenon of increasing blastemal predominance with passaging of the xenografts.

2. If there is a comparison between the molecular findings and the histology between the xenografts and the primary tumor the following question needs to be answered Is the histology of the primary tumor taken from the same specimen that was used for the xenograft? In other words a biopsy was taken from the primary tumor for the xenograft and was this biopsy splitted into 2 parts one for the xenograft and the other one for the histology and molecular findings? Is this meant by tumor pairs? If this is not the case, can this be a reason for discrepancies between histologies, finding diffuse anaplasia in the primary tumor but not in the xenograft? This question is also important related to line 405 - 408. For the neoadjuvant treated tumors and the discrepancy between primary tumor and xenograft the question would be is the xenograft showing more malignant clones, meaning blastemal subtypes or anaplasia?

Response: The histology of the primary tumor was derived from review of all surgical pathology tissue blocks archived at the time of the original patient tumor resection and reflects the summary of findings for the entire tumor, as would be clinically reported. All pathology was re-reviewed and confirmed for the current study. The specimen taken for xenografting was not exactly matched with a specific tissue block from the tumor resection. For analysis of the xenografts, a tissue specimen was split into two pieces – one for molecular analysis and one for histology and immunohistochemistry. In the discussion section, we note that histologic differences between the primary tumor and xenograft could either be due to primary tumor heterogeneity or clonal selection/evolution in the process of xenograft establishment. We have performed a subclonal analysis using Target capture/deep sequencing of the genetic variants explored in our study to compare clonality of the primary tumor to the xenograft.

Minor

point:

line 124: after neoadjuvant chemotherapy high risk blastemal tumors are described as blastemal subtypes. Blastemal predominant tumors are only named in this way after primary surgery, to distinguish them from eachother. This is of relevance, as in line 142, it is unclear, if this blastemal predominant tumor was primarily operated or not.

Response: In this context, blastemal predominant is simply a description of the xenograft histology and does not refer to primary surgery or risk stratification. For the primary tumors, blastemal predominance or diffuse anaplasia after neoadjuvant treatment is indicated in Figure 1 – SIOP Post-treatment histology – high risk. This is described in the legend to Figure 1.

Reviewers' comments:

Reviewer #1 (Remarks to the Author):

The manuscript has improved. Authors have extended the genetic analysis of these WT Xn. The manuscript is still descriptive but 45 tumors including anaplastic tumors are a large cohort and can contribute to the understanding of WT biology and perhaps serve as a platform for future drug screens that may help others. Therefore, I am overall more positive.

Authors must include the passage number at which they performed RNAseq and all other analysis for each xenograft (since that can have a large effect on its population repertoire). This is a critical issue and should be carefully detailed.

Unfortunately, authors still fail to cite critical literature including studies that analyzed global gene expression in WT-PDX as well as the nephric progenitor genes in WT-PDX (refs: Cancer Res. 2006 Jun 15;66(12):6040-9, Stem Cells. 2008 Jul;26(7):1808-17). This current study is indeed comprehensive but largely recapitulates and validates older findings in a broader manner. This should be clearly put forward.

Moreover, the all idea of earlier WT-PDX studies is based on the finding that human WT blastema that harbors the cancer initiating/stem cells engrafts and propagates in the mouse. This important property (that is validated here by PDX RNA seq) previously allowed the identification and characterization of CIC/CSC in WT-PDX and their therapeutic targeting and clinical relevance. Thus, WT-PDX have been used in the past for studying the biology of WT and for analyzing the effects of novel drugs based on blastema propagation in the PDX model. (refs: EMBO Mol Med. 2013 Jan;5(1):18-37, Mol Cancer Ther. 2017 Nov;16(11):2462-2472). This should be all included.

Reviewer #2 (Remarks to the Author):

1. This is a greatly improved manuscript though it remains a tragedy that the tumours were only characterised by exome sequencing, fundamentally limiting the phylogenetic insight that can be gained.
2. Can the authors please provide a table of all the somatic mutations they found which they consider to be real (i.e. which passed their filtering)? This needs to be published alongside the paper. I would like to see this data to get a sense of the quality of their mutation calling. (If they were included already, I might not have been able to see them as some of the XLS files were corrupted).
3. The manuscript would benefit from a basic characterisation of the exome data, i.e. number of subs and indels per tumour at the very least (could be incorporated into Figure 1).

Reviewer #3 (Remarks to the Author):

This is the revised version of a manuscript dealing with the biological heterogeneity of Wilms tumor in 45 patient derived xenografts. The authors provide an important scientific resource for researchers dealing with translational medicine in Wilms tumor. The revised version of the manuscript has significantly improved.

Nevertheless one question of reviewer 3 cannot be answered by the manuscript as the methodology is not allowing it. A biopsy that was used for the xenografts was not splitted into 2 parts to compare them. This would be needed in an upcoming research to demonstrate the intratumor heterogeneity by using different biopsies from the same tumor of a patient. In bilateral cases it would also be mandatory to get biopsies from both sides, as they may be different in histology etc. Such an approach is important to see if response to drugs is the same for different xenografts from one patient. Only if that is the case this will have an impact in selecting the right treatment for a patient. In preoperative chemotherapy as used in SIOP we have learned that part of blastema is getting regressive whereas another part remains blastemal being treatment resistant. This happens in the same tumor of a patient.

One minor finding please add in the abstract the number of the patients and not only writing: 'These WTPDX include 6 from patients with diffuse anaplastic tumors, 9 from patients who experienced disease relapse, and 13 from patients with bilateral disease.' If you would count these numbers you come to 28. You need to write: 'out of 45 ...'.

Another point deals with figure 1: Here you have in the clinical characteristics stage, where you will find the 13 patients with stage V (bilateral). The question is, if none of the 13 patients had metastatic disease? Because stage V can also be stage IV. To have this in one line you should add in the legend that none of stage V patients had metastatic disease. If one bilateral case had also metastatic disease, you need to mention this as well.

Re: Second revision of manuscript (NCOMMS-18-04822) to *Nature Communications: Forty-five patient derived xenografts capture the clinical and biological heterogeneity of Wilms tumor*

Dear Reviewers:

After completing all the requested revisions based on reviewer comments to our resubmitted manuscript, I am pleased to submit the second revision of the above-titled manuscript to be considered for publication as an *Article* in *Nature Communications*.

This manuscript has not been published in part or in entirety and is not under consideration by another journal. All studies were conducted under the appropriate institutional review board and animal care and use committee-approved protocols. All of the authors have approved the manuscript and agree with submission to your journal. There are no conflicts of interest to declare. Genome-wide data have been submitted to and accepted by the European Genome-phenome Archive (EGA) and Gene Expression Omnibus (GEO) databases as outlined in the data availability statement and reporting summary.

We believe that this freely available, important scientific resource warrants publication in *Nature Communications* because it stands to significantly accelerate research for Wilms tumor patient groups with suboptimal outcomes if it reaches a broad audience. We believe this publication represents a resource of significant importance to specialists in the field of translational pediatric oncology research and scientists who study pediatric renal tumors. We have worked extremely hard to complete these revisions to your satisfaction because we believe in the importance of this resource, which took 10 years to assemble at a center dedicated to pediatric oncology research.

Please find our itemized responses to the reviewer comments below.

Sincerely,

Andrew Jackson Murphy, MD
Assistant Member, Department of Surgery
St. Jude Children's Research Hospital

Reviewers' comments:

Reviewer #1 (Remarks to the Author):

The manuscript has improved. Authors have extended the genetic analysis of these WT Xn. The manuscript is still descriptive but 45 tumors including anaplastic tumors are a large cohort and can contribute to the understanding of WT biology and perhaps serve as a platform for future drug screens that may help others. Therefore, I am overall more positive.

Response: Thank you for your review of our manuscript.

Authors must include the passage number at which they performed RNAseq and all other analysis for each xenograft (since that can have a large effect on its population repertoire). This is a critical issue and should be carefully detailed.

Response: The passage number for all xenografts used in the analysis is now indicated in the results section of the manuscript: “Second passage (P2) WTPDX material was used for all molecular analyses except P1 was used for KT-20 and P3 for KT-64 due to specimen availability.”

Unfortunately, authors still fail to cite critical literature including studies that analyzed global gene expression in WT-PDX as well as the nephric progenitor genes in WT-PDX (refs: Cancer Res. 2006 Jun 15;66(12):6040-9, Stem Cells. 2008 Jul;26(7):1808-17). This current study is indeed comprehensive but largely recapitulates and validates older findings in a broader manner. This should be clearly put forward.

Response: The above landmark studies using Wilms tumor xenografts have been referenced and discussed in the manuscript, making it clear that our findings validate and extend prior observations in a larger number of xenograft models that represent the spectrum of Wilms tumor.

Moreover, the all idea of earlier WT-PDX studies is based on the finding that human WT blastema that harbors the cancer initiating/stem cells engrafts and propagates in the mouse. This important property (that is validated here by PDX RNA seq) previously allowed the identification and characterization of CIC/CSC in WT-PDX and their therapeutic targeting and clinical relevance. Thus, WT-PDX have been used in the past for studying the biology of WT and for analyzing the effects of novel drugs based on blastema propagation in the PDX model. (refs: EMBO Mol Med. 2013 Jan;5(1):18-37, Mol Cancer Ther. 2017 Nov;16(11):2462-2472). This should be all included.

Response: The above landmark studies using Wilms tumor xenografts have been referenced and discussed in the manuscript. Thank you for bringing these studies to our attention.

Reviewer #2 (Remarks to the Author):

1. *This is a greatly improved manuscript though it remains a tragedy that the tumours were only characterised by exome sequencing, fundamentally limiting the phylogenetic insight that can be gained.*

Response: Although we thought whole genome sequencing was outside the scope of the current project, we are strongly considering this method for future studies.

2. *Can the authors please provide a table of all the somatic mutations they found which they consider to be real (i.e. which passed their filtering)? This needs to be published alongside the paper. I would like to see this data to get a sense of the quality of their mutation calling. (If they were included already, I might not have been able to see them as some of the XLS files were corrupted).*

Response: This information is included in supplementary data 3.

3. *The manuscript would benefit from a basic characterisation of the exome data, i.e. number of subs and indels per tumour at the very least (could be incorporated into Figure 1).*

Response: This summary information has been added to supplementary data 3.

Reviewer #3 (Remarks to the Author):

This is the revised version of a manuscript dealing with the biological heterogeneity of Wilms tumor in 45 patient derived xenografts. The authors provide an important scientific resource for researchers dealing with translational medicine in Wilms tumor. The revised version of the manuscript has significantly improved.

Response: Thank you for your review of our manuscript.

Nevertheless one question of reviewer 3 cannot be answered by the manuscript as the methodology is not allowing it. A biopsy that was used for the xenografts was not splitted into 2 parts to compare them. This would be needed in an upcoming research to demonstrate the intratumor heterogeneity by using different biopsies from the same tumor of a patient. In bilateral cases it would also be mandatory to get biopsies from both sides, as they may be different in histology etc. Such an approach is important to see if response to drugs is the same for different xenografts from one patient. Only if that is the case this will have an impact in selecting the right treatment for a patient. In preoperative chemotherapy as used in SIOP we have learned that part of blastema is getting regressive wherease another part remains blastemal being treatment resistant. This happens in the same tumor of a patient.

Response: The effect of sampling phenomenon in the setting of significant intratumor heterogeneity is discussed in the discussion section of the manuscript. Based on the reviewer comments, we have started to establish xenografts incorporating a variety of areas from the same tumor, or both tumors in the case of bilateral Wilms tumor, in preparation for future research projects.

One minor finding please add in the abstract the number of the patients and not only writing: 'These WTPDX include 6 from patients with diffuse anaplastic tumors, 9 from patients who experienced disease relapse, and 13 from patients with bilateral disease.' If you would count these numbers you come to 28. You need to write: 'out of 45 ...'.

Response: The language in the abstract has been rewritten to clarify this point: ***“Among these 45 total WTPDX, 6 from patients with diffuse anaplastic tumors, 9 from patients who experienced disease relapse, and 13 from patients with bilateral disease are included.”***

Another point deals with figure 1: Here you have in the clinical characteristics stage, where you will find the 13 patients with stage V (bilateral). The question is, if none of the 13 patients had metastatic disease? Because stage V can also be stage IV. To have this in one line you should add in the legend that none of stage V patients had metastatic disease. If one bilateral case had also metastatic disease, you need to mention this as well.

Response: 4 of the 13 patients with bilateral disease also had metastases. The staging in Figure 1 and the accompanying legend have been updated to reflect this: ***“Children’s Oncology Group (COG) disease stage is indicated. For bilateral WT cases (stage 5) also with metastasis present, the stage is indicated as 5/4.”***

Reviewers' comments:

Reviewer #2 (Remarks to the Author):

Alas, Suppl. Table 3 does not contain the data I asked for. The authors need to publish a list of ALL mutations that they consider to be somatic, substitutions and indels, synonymous and non-synonymous. Could the authors provide this so that I can assess the data?

Reviewer #3 (Remarks to the Author):

All remarks of the reviewers are sufficiently addressed. I haave no further comments to the authors. In my view the paper can be published.

Re: Third revision of manuscript (NCOMMS-18-04822) to Nature Communications: Forty-five patient derived xenografts capture the clinical and biological heterogeneity of Wilms tumor

Dear Reviewers:

After completing all the requested revisions based on reviewer comments to our resubmitted manuscript, I am pleased to submit the third revision of the above-titled manuscript to be considered for publication as an Article in Nature Communications.

This manuscript has not been published in part or in entirety and is not under consideration by another journal. All studies were conducted under the appropriate institutional review board and animal care and use committee-approved protocols. All of the authors have approved the manuscript and agree with submission to your journal. There are no conflicts of interest to declare. Genome-wide data have been submitted to and accepted by the European Genome-phenome Archive (EGA) and Gene Expression Omnibus (GEO) databases as outlined in the data availability statement and reporting summary.

We believe that this freely available, important scientific resource warrants publication in Nature Communications because it stands to significantly accelerate research for Wilms tumor patient groups with suboptimal outcomes if it reaches a broad audience. We believe this publication represents a resource of significant importance to specialists in the field of translational pediatric oncology research and scientists who study pediatric renal tumors. We have worked extremely hard to complete these revisions to your satisfaction because we believe in the importance of this resource, which took 10 years to assemble at a center dedicated to pediatric oncology research.

Please find our itemized responses to the reviewer comments below.

Reviewers' comments:

Reviewer #2 (Remarks to the Author):

Alas, Suppl. Table 3 does not contain the data I asked for. The authors need to publish a list of ALL mutations that they consider to be somatic, substitutions and indels, synonymous and non-synonymous. Could the authors provide this so that I can assess the data?

Response: Supplementary Data file 3 has been updated to include all somatic silent single nucleotide variants, non-silent variants, and indels.

Reviewer #3 (Remarks to the Author):

All remarks of the reviewers are sufficiently addressed. I have no further comments to the authors. In my view the paper can be published.

Response: Thank you for your review of our manuscript.

REVIEWERS' COMMENTS:

Reviewer #2 (Remarks to the Author):

I have no further comments.

October 31, 2019

Nature Communications
Editorial and Production Offices
Suite 4500, One New York Plaza
New York, NY 10004
naturecommunications@nature.com

Re: Final submission of manuscript (NCOMMS-18-04822) to *Nature Communications*: *Forty-five patient derived xenografts capture the clinical and biological heterogeneity of Wilms tumor*

Dear Reviewers:

I am pleased to submit the final version of the above-titled manuscript for publication as an *Article* in *Nature Communications*.

This manuscript has not been published in part or in entirety and is not under consideration by another journal. All studies were conducted under the appropriate institutional review board and animal care and use committee-approved protocols. All of the authors have approved the manuscript and agree with submission to your journal. There are no conflicts of interest to declare. Genome-wide data have been submitted to and accepted by the European Genome-phenome Archive (EGA) and Gene Expression Omnibus (GEO) databases as outlined in the data availability statement and reporting summary.

We believe that this freely available, important scientific resource warrants publication in *Nature Communications* because it stands to significantly accelerate research for Wilms tumor patient groups with suboptimal outcomes if it reaches a broad audience. We believe this publication represents a resource of significant importance to specialists in the field of translational pediatric oncology research and scientists who study pediatric renal tumors. We have worked extremely hard to complete these revisions to your satisfaction because we believe in the importance of this resource, which took 10 years to assemble at a center dedicated to pediatric oncology research.

Please find our itemized responses to the reviewer comments below.

Sincerely,

Andrew Jackson Murphy, MD
Assistant Member, Department of Surgery
St. Jude Children's Research Hospital

Reviewers' comments:

Reviewer #2 (Remarks to the Author):

I have no further comments.

Response: Thank you for your review of our manuscript.